# Multitasking Models are Robust to Structural Failure: A Neural Model for Bilingual Cognitive Reserve

**Giannis Daras***
The University of Texas at Austin
giannisdaras@utexas.edu

**Negin Raoof***
The University of Texas at Austin
neginraoof@gmail.com

**Zoi Gkalitsiou**
The University of Texas at Austin
zoi.gkalitsiou@austin.utexas.edu

**Alexandros G. Dimakis**
The University of Texas at Austin
dimakis@austin.utexas.edu

## Abstract

We find a surprising connection between multitask learning and robustness to neuron failures. Our experiments show that bilingual language models retain higher performance under various neuron perturbations, such as random deletions, magnitude pruning and weight noise compared to equivalent monolingual ones. We provide a theoretical justification of this robustness by mathematically analyzing linear representation learning and showing that multitasking creates more robust representations. Our analysis connects robustness to spectral properties of the learned representation and proves that multitasking leads to higher robustness for diverse task vectors. We open-source our code and models in the following URL: https://github.com/giannisdaras/multilingual_robustness.

## 1 Introduction

Converging evidence from cognitive science research indicates that bilingualism increases brain robustness by reducing the rate of cognitive decline due to aging [1, 2] and delaying the onset of symptoms of dementia [3, 4]. It appears that individuals who speak more than one language on a regular basis are able to maintain typical cognitive functioning despite neural degeneration. This mismatch between cognitive functioning and brain pathology is called Cognitive Reserve [5], and its underlying mechanisms are poorly understood and are an active topic of investigation.

Inspired by this research, we study whether *artificial* neural networks are more robust when trained on multiple languages or multiple tasks. Our experiments demonstrate that training on multiple tasks indeed increases structural robustness. We train monolingual and bilingual GPT-2 models with the same architecture and dataset sizes. Initially, monolingual GPT-2 [6] models are slightly outperforming the bilingual ones, but when we introduce structural noise (by randomly deleting neurons or adding noise to the weights) bilingual models degrade more gracefully and eventually outperform the monolingual models in the high-noise regime. For some amount of noise, bilingual models start outperforming the monolingual ones demonstrating a *cross-over* in performance due to their increased robustness. We observe this phenomenon for numerous models across three different types of corruption: additive Gaussian noise to the weights, random weight pruning and magnitude-based weight pruning [7].

**Our Contributions:** We provide a theoretical justification of this phenomenon by mathematically analyzing linear multitask representation learning [8, 9]. Our analysis shows that introducing more

---

*equal contribution.

36th Conference on Neural Information Processing Systems (NeurIPS 2022).

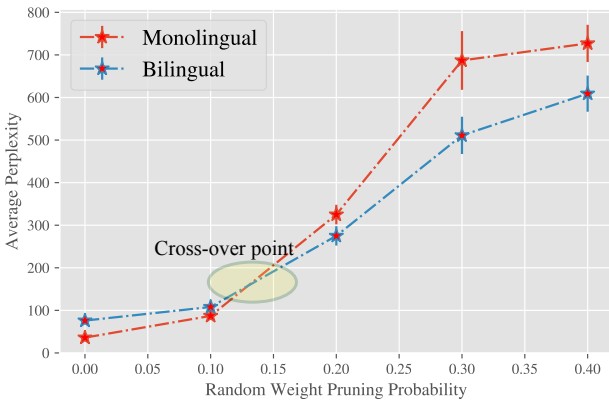

Figure 1: Performance of monolingual and bilingual GPT-2 models with the same architecture and training dataset size. We show the performance as we randomly erase weights. The x-axis indicates the probability of erasing an attention weight parameter (setting to it zero). The y-axis indicates the average perplexity over 20 runs with 95% confidence intervals. The bilingual model initially shows slightly worse performance, but as more weights are deleted, the monolingual model declines faster and performs worse in the highly damaged regime. This indicates that the bilingual GPT-2 model is more robust to neuron weight erasures. We show similar results for several models and types of errors in our experimental section.

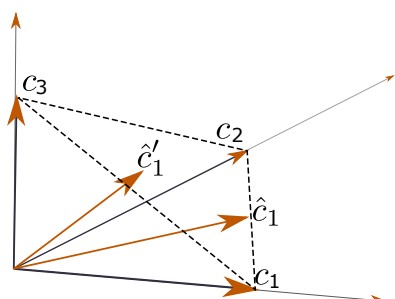

Figure 2: Let $c_1, c_2, c_3$ be the standard basis of $\mathbb{R}^3$. For two tasks, the best one dimensional approximation to $c_1, c_2$ is $\hat{c}_1 = [1/2, 1/2, 0]^T$ but the best one dimensional approximation to three tasks $c_1, c_2, c_3$ is $\hat{c}'_1 = [1/3, 1/3, 1/3]^T$. Multi-tasking is creating $\ell_2$ regularization since $||\hat{c}'_1||_2 < ||\hat{c}_1||_2$. It is important that the original task vectors $c_1, c_2, c_3$ are orthogonal i.e. diverse, since this creates regularization.

diverse tasks creates $\ell_2$ regularization in the linear task heads. Further, we formally connect the Euclidean norm of the learned representations to structural robustness under errors in the network weights. Our main theorem establishes that multitasking leads to higher robustness to additive noise for linear representations when the task vectors are selected as random and independent Gaussian vectors. Our results also establish that when the tasks are significantly overlapping, multitasking does not lead to higher robustness and hence task diversity is necessary.

We experimentally observe that multitasking increases structural robustness for numerous networks and multiple problems including MNIST, CIFAR10, Newsgroup20, GPT models and finetuned GPT models on GLUE tasks. We train networks under exactly comparable dataset and architecture conditions and show that models become more robust to structural failures as they are trained with more tasks. We experiment with three different types of structural failures and show robustness increases for all of them. We also experimentally observe that the addition of diverse tasks seems to regularize the model weights, as we predict in our theoretical analysis.

## 2   Theoretical Analysis

**Building intuition.** We start with a small numerical example to build intuition. Given a feature vector $x \in \mathbb{R}^d$ we compute a $k$ dimensional linear representation $Wx$ using a matrix $W \in \mathbb{R}^{k \times d}$. We choose

$W$ such that we best approximate a set of ground truth task vectors, $\{c_1, c_2, ..., c_T\}$, that lie in $\mathbb{R}^d$. The learned approximation is $\hat{c}_i = W^T \gamma_i$. Essentially, we use linear combinations of the columns of $W^T$ to approximate the task vectors. For simplicity, we assume that the columns of $W^T$ are unit norm. We study the case where $k < T$, otherwise there are infinite solutions.

Assume we work in $d = 3$ dimensions with $T = 3$ total tasks, $c_1 = [1, 0, 0]^T$, $c_2 = [0, 1, 0]^T$, $c_3 = [0, 0, 1]^T$. Set our learned representation dimension to be $k = 1$ dimensional. When $T = 2$, using only the first two tasks $c_1, c_2$, an optimal solution is $W = \frac{1}{\sqrt{2}}[1, 1, 0]$. The corresponding linear head is now the scalar $\gamma_1 = \frac{1}{\sqrt{2}} = \gamma_2$ and the approximate vectors are $\hat{c}_1 = W^T \gamma_1 = [0.5, 0.5, 0]^T = \hat{c}_2$. Therefore the best one dimensional subspace to jointly approximate $c_1, c_2$ is the span of $W = \frac{1}{\sqrt{2}}[1, 1, 0]$. Now we introduce one more task and find the one dimensional subspace that best approximates $c_1, c_2, c_3$. That becomes $W' = \frac{1}{\sqrt{3}}[1, 1, 1]$ with linear heads $\gamma'_1 = \frac{1}{\sqrt{3}} = \gamma'_2 = \gamma'_3$. The approximate vectors now are $\hat{c}'_1 = (W')^T \gamma'_1 = [1/3, 1/3, 1/3]^T = \hat{c}'_2 = \hat{c}'_3$. Notice that $||\hat{c}'_i||^2 = 1/3$ for 3 tasks but $||\hat{c}_i||^2 = 1/2$ for two tasks. The point is that for *more tasks, the vector that jointly approximates all task vectors becomes shorter.* Equivalently, the $\ell_2$ norm of the linear task heads *decreases* from $\gamma_i = \frac{1}{\sqrt{2}}$ to $\gamma'_i = \frac{1}{\sqrt{3}}$ as the tasks increased from two to three showing how multitasking creates regularization. A graphical representation of this example is given in Figure 2. It is important that the task vectors $c_i$ are orthogonal, increasing the effective dimensionality of the problem. The intuition is that diverse tasks increase the effective dimension, making the best approximation vector shorter.

Our main theoretical result is that this phenomenon is quite general and makes multitasking lead to structural robustness. We connect the norm of the approximated task vectors with robustness to weight perturbations and show that for Gaussian, independent task vectors the average norm shrinks as more tasks are added. This is intuitive since high dimensional Gaussian vectors are near-orthogonal. Surprisingly, we empirically show that task vectors for numerous problems also exhibit this behavior.

**Analysis.** We consider a neural network $f_\theta : \mathbb{R}^d \to \mathbb{R}^k$ and a collection of tasks $\{\mathcal{T}_1, ..., \mathcal{T}_T\}$. We are trying to learn $\theta, \gamma_i \in \mathbb{R}^k$ to solve the following optimization problem:

$$\operatorname{argmin}_{\theta, \{\gamma_1, ..., \gamma_T\}} \sum_{i=1}^{T} \mathbb{E}_{(x,y) \in \mathcal{T}_i} \mathcal{L}(\gamma_i^T f_\theta(x), y). \tag{1}$$

The neural network $f_\theta$ can be as simple as a single matrix $W : \mathbb{R}^d \to \mathbb{R}^k$. For linear networks, we consider the following dataset generation process: for task $\mathcal{T}_i$, we sample a Gaussian $x$ and we generate its label $y$ by taking the inner-product with a task vector $c_i$, i.e. $y = c_i^T x$ for task $\mathcal{T}_i$. Given infinite samples and MSE loss, the optimization problem of (1) is equivalent to the following problem.

**Definition 2.1** (Optimization Problem)**.** Let $k < T < d$. We define the Factorized Best Rank-$k$ approximation of a matrix $C \in \mathbb{R}^{d \times T}$ as the optimization problem:

$$W^*, \Gamma^* = \operatorname{argmin}_{W \in \mathbb{R}^{k \times d}, \Gamma \in \mathbb{R}^{k \times T}} \left\| W^T \Gamma - C \right\|_F^2. \tag{2}$$

We are interested in the case when the dimensionality of the representation $k$ is smaller than the number of tasks $T$, otherwise the best Rank-$k$ approximation of $C$ is not unique.

The following Proposition states that in the considered setting, Problem 2 can be solved with SVD.

**Proposition 2.2.** *For any matrix $C \in \mathbb{R}^{d \times T}$ with distinct singular values, any solution of 2.1 satisfies:*

$$W^{*T} \Gamma^* = U \Sigma_k V^T, \tag{3}$$

*where $U \Sigma V^T$ is the SVD of $C$ and $\Sigma_k$ is the same as $\Sigma$ except than the last $T - k$ diagonal entries that are zeroed out.*

The fact that the Singular Value decomposition computes the best rank-$k$ approximation to a matrix can be found in several textbooks e.g. Golub and Van Loan [10], Blum et al. [11].

This proposition establishes that $W^* = U^T$ and $\Gamma^* = \Sigma_k V^T$ is a valid solution of (2). Onwards, we will be calling this the SVD Solution.

**Definition 2.3.** We define the SVD solution of (2), to be:

$$W_{\text{SVD}} = U^T, \quad \Gamma_{\text{SVD}} = \Sigma_k V^T. \tag{4}$$

We note that if any multitask learning algorithm is used to obtain $W^*, \Gamma^*$, one can run Gram-Schmidt to make $W^*$ orthonormal and hence obtain the factorization we use. It is important that $W$ stays normalized and all scaling is pushed to $\Gamma$ since to measure robustness to weight shifts, we are going to add noise to $W$ only, and higher $W$ scaling is equivalent to lower effective noise.

We study how the performance is affected when the representation network, $f_\theta$, is corrupted.

**Definition 2.4.** For any sample $x$, the **Mean Squared Error (MSE)** for task $i$ is defined to be the expected error between the model prediction under noise and the true value $y$. Namely,

$$\text{MSE}^i = \mathbb{E}_{\theta_c} \left[ (\gamma_i^T f_{\theta_c}(x) - y)^2 \right], \tag{5}$$

where $f_{\theta_c}$ is the model that emerges after corrupting $f_\theta$.

This measures how well the model approximates the ground truth under the presence of noise and under the constraint of a joint representation for multiple tasks.

The simplest corruption process to study is adding noise to the representation matrix, i.e.

$$W_c = W + N, \quad N_{ij} \sim \mathcal{N}(0, \sigma^2), \text{ i.i.d} \tag{6}$$

Then, we denote the mean squared error for the task $i$ with $\text{MSE}^{i,\sigma^2}$ and the average mean squared error across the $T$ tasks with $\overline{\text{MSE}}^{T,\sigma^2}$. We are now ready to introduce our results.

**Theorem 2.5** (Mean Squared Error for Additive Noise). *Let $C \in \mathbb{R}^{d \times T}$ be a matrix with distinct singular values $\sigma_1 > \sigma_2 > ... > \sigma_T$. Let $W, \Gamma$ be the SVD solution of* (2). *Under the Additive Noise Model defined in* (6), *we have that:*

*Average MSE without noise*

$$\overline{\text{MSE}}^{T,\sigma^2} = \overline{\text{MSE}}^{T,0} + \frac{\sum_{i=1}^{k} \sigma_i(C)^2}{T} \cdot \sigma^2 \,. \tag{7}$$

*Average MSE under noise*     *Noise Variance*

As shown, the noisy MSE decomposes into the sum of the noiseless MSE plus the noise variance times a function that depends on the number of tasks:

$$R(T) = \frac{\sum_{i=1}^{k} \sigma_i(C)^2}{T}. \tag{8}$$

It is important to emphasize that as more tasks are added, the matrix $C$ changes, but the interlacing theorem allows us to connect the singular values of smaller submatrices, as discussed in the Appendix. $R(T)$ is the robustness slope: if a model with $T$ tasks has smaller slope, it will eventually outperform a model with, say $T-1$ tasks and larger slope, for sufficiently large noise. This is true even if the noiseless performance for the $T-1$-task model is better, indicating a cross-over in MSE. Therefore the key is understanding when the sum of the top $k$ singular values of $C$ scales sublinearly in $T$. This is not true for tasks that are aligned, but we can show it holds for independent Gaussian task vectors. We believe it holds for more general families of diverse task vectors and our experiments verify it also holds for numerous real task vectors learned from text and vision datasets.

**Connection with $l_2$ regularization.** For the SVD solution (see Definition 4), the sum of the top-k singular values squared is the squared Frobenius norm of $\Gamma$. Indeed, we have that $||\Gamma_{\text{SVD}}||_F^2 = ||\Sigma_k V^T||_F^2$. Since $\Sigma_k$ is a diagonal matrix, each row of $\Sigma_k V^T$ is a rescaling of the corresponding row of $V^T$. Rows of $V^T$ have norm 1, hence the i-th row of $\Sigma_k V^T$ will have norm $\sigma_i$. The Frobenius norm squared is just the sum of the squared norms of the rows. Hence, we get that

$$||\Gamma_{\text{SVD}}||_F^2 = \sum_{i=1}^{k} \sigma_i(C)^2. \tag{9}$$

Using this simple observation, we can get the following alternative expression of Theorem 2.5.

**Corollary 2.6.** *Let $C \in \mathbb{R}^{d \times T}$ be a matrix with distinct singular values. Let $W, \Gamma$ be the SVD solution of* (2). *Under the Additive Noise Model defined in* (6), *we have that:*

$$\overline{\text{MSE}}^{T,\sigma^2} = \overline{\text{MSE}}^{T,0} + \frac{||\Gamma||_F^2}{T} \sigma^2 \,. \tag{10}$$

Corollary 2.6 provides two important insights: i) the normalization with the number of tasks that appears in (7) is justified since the Frobenius norm of $\Gamma$ grows with the number of task, ii) if we can prove that the slope (defined in Equation (8)) is dropping, then we are effectively proving that multitasking gives $l_2$ regularization as we showed in the toy introductory example. This also holds for the case of Gaussian, i.i.d. task vectors, as shown in the following theorem.

**Theorem 2.7.** *Let $C \in \mathbb{R}^{d \times T}$ be a random matrix with Gaussian, i.i.d. entries of variance $1/d$ and $d = \Omega(T^3)$. Let $C_t, C_{t+1}$ be the matrices formed by selecting the first $t, (t+1)$ columns of $C$. Then, there is a noise level $\sigma_{\text{thres}}$ such that with probability $\geqslant 1 - \exp\left(-\Omega\left(\sqrt{d}\right)\right)$, the SVD solutions (see (4)) of (2) (for $C_t, C_{t+1}$ respectively), under the noise corruption model, satisfy:*

$$\overline{\mathrm{MSE}}^{t+1,\sigma^2} < \overline{\mathrm{MSE}}^{t,\sigma^2}, \quad \forall\, \sigma \geqslant \sigma_{\text{thres}}. \tag{11}$$

*Remark* 2.8. In words, this result shows that adding new tasks gives **provably** increased robustness to high noise corruption in the weights, when the task vectors are Gaussian.

*Remark* 2.9. Observe that the MSE under noise drops for *every single new task added*. The assumption $d = \Omega(T^3)$, can be relaxed to $d = \Omega(t^3)$, and we get increased robustness for the first $t$ added tasks. Nevertheless, for most applications $d = \Omega(T^3)$ is a realistic assumption: Even for our smallest dataset MNIST $d = 728$, and we experiment with up to 10 tasks.

## 3 Experimental Evaluation

We divide the experimental section in two parts. In the first part, we add noise to the final linear representation layer of various networks and verify that our theoretical analysis agrees with experimentally observed multitasking robustness on real datasets (MNIST, CIFAR10, NewsGroup20). In the second part, we show that multitasking leads to robustness to general weight corruptions in any layer of a complex transformer. Specifically, we show that multilingual Language Models are more robust to weight shifts (across all the layers) compared to monolingual trained under the same setting. This is the first evidence of increased Cognitive Reserve in bilingual artificial neural networks.

**Experiments with Linear Representation Layers.** We perform experiments on three datasets (MNIST, CIFAR10, Newsgroup20) and two modalities (Vision and Language). The datasets normally involve one classification task each. We create multiple binary tasks by distinguishing between pairs of labels. For example, in CIFAR10, one task might be to distinguish between dogs and cats and another between airplanes and cars. We assign a value in $[0, 1]$ to each sample for each task to transform them to regression tasks (to match our theory). For example, if task $i$ is to distinguish between dogs and cats, value 0 corresponds to dog and value 1 to cat.

The second issue is learning the task vectors from training data. For MNIST, we can simply learn a linear layer $C$ with columns $\{c_1, ..., c_T\}$ such that: $c_i^T x \approx y$ for each task. For more complex datasets like CIFAR or Newsgroup20, linear networks have lower performance and hence it is less interesting to examine their robustness. Instead, we first use another network to extract representations $g_\theta(x)$ and then learn a linear layer acting on the encodings such that $c_i^T g_\theta(x) \approx y$. For CIFAR we used a pre-trained Resnet50 as the encoder while for NewsGroup, a pre-trained BERT [12]. We would like to point out that our theory is still valid for this case – this is equivalent to the linear layer $C$ receiving inputs from a learned representation as opposed to the features directly. As the number of tasks increase, we reduce the number of training examples per task. We do this to make sure that the total training dataset size stays the same as the number of tasks increase.

Figure 3 shows how the average MSE behaves as noise increases for different number of tasks. Note that even though all models begin from roughly the same performance in the noiseless setting, the multitask models are much more robust to the corruption of their weights consistently among all the datasets and modalities. This is aligned with our theoretical analysis which predicts that the robustness slope (defined in Equation (8)) decreases with the number of tasks. We calculate robustness slopes for learned task vectors for real datasets and plot their decay in the Appendix, where we further include all the details of how these models were trained.

**Experiments with Language Models.** Our objective is to compare robustness to neural weight perturbations in monolingual and bilingual language models. We use the following perturbation

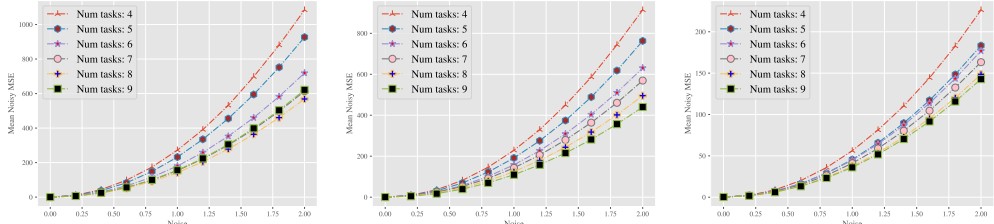

Figure 3: MSE of model (versus optimal task vector) as a function of noise added to the weights. From left to right: MNIST, CIFAR10, NewsGroup20. As shown for all these datasets, adding tasks increases the robustness of the model to noise in the weights.

models: 1) Random deletion of weight parameters: we zero-out $p$ percent of the attention layer weights, 2) Magnitude pruning: we sort model attention weights by the magnitude and delete the smallest p percent of weights [7], 3) Random normal noise: we add zero-mean random Gaussian noise with standard deviation $\sigma^2$ to the attention weights.

On the selection of the linguistic pair, we selected Greek, a highly inflected language with very different morphology, syntax and phonology compared to English. It also uses a different script since Greek characters were not Romanized. This minimizes transfer between languages, something we wanted to avoid. In the Appendix, we present additional experiments for other Romance languages.

The dataset for the bilingual model is a concatenation of articles from English and Greek Wikipedia. To avoid the computational cost of training for a new language, we start from the pre-trained GPT-2 (small)[6] and we use the Language Model Recycling Technique, introduced in [13]. GPT-2 small is a transformer-based architecture for causal language modeling, with 12 attention blocks and 124M parameters. The tokenizer uses Byte Pair Encoding and has a vocabulary of $50,257$ tokens. For the bilingual model, we generate a new tokenizer, vocabulary and embedding layer without changing the architecture. We keep the vocabulary size the same, as changing the vocabulary size can affect the scale of the perplexity score for these models. Note that Wikipedia documents were not in the original training of GPT-2, but our monolingual baseline was subsequently finetuned on English Wikipedia. Details on all our training hyperparameters are included in the Appendix.

We measure the quality of generated text using perplexity. Our bilingual model achieves 89 perplexity on a randomly picked subset of the OSCAR [14] dataset and 76 perplexity on the English IMDB dataset [15]. Monolingual GPT-2 model achieves 36 perplexity on the IMDB dataset. In the Appendix we include generated text for both the models. Although the perplexity of the bilingual model does not match the pre-trained GPT-2, the generated text is of reasonable quality text in both languages.

**Text Generation.** Our first experiment is to compare the performance of both models under various parameter perturbations. First, we try deleting a random portion $p$ ($p$ from $0\%$ to $40\%$) of attention layers' weight to observe and compare the trend of decay in text generation quality between the two models. We evaluate both models on the IMDB dataset. As the graph in Figure 1 shows, the monolingual model starts with text predictions closer to the source text, resulting in lower perplexity without noise. However, as we delete a more significant portion of weights, the bilingual model matches the performance of the monolingual one and eventually outperforms that.

Next, we try magnitude-based pruning of a portion of weights, $p$, to observe and compare the trend of decay in text generation quality between the two models. We sort the attention layer weights by the magnitude and set $p$ percent of weights with the lowest magnitude to zero. Again, we use the IMDB dataset to evaluate models. The graphs in Figure 4 show that as the training process continues, the model achieves a lower perplexity. Moreover, pruning additional weights has a less substantial impact on the model's performance. This graph shows that training the pre-trained GPT-2 model for a few epochs on a bilingual dataset significantly improves robustness to weight perturbations.

In another experiment, we observe how the maximum singular value of the weight matrices changes throughout training process. We track the maximum singular value of attention layer weights. We use a pretrained GPT-2 model baseline, and train this model for 16k iterations on English text data from Wikipedia. Resuming from this checkpoint, we train two new models: 1) We continue training model

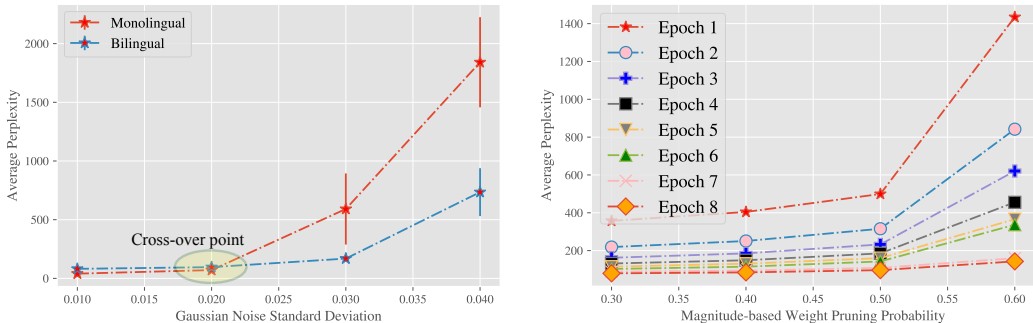

Figure 4: Robustness to magnitude-based weight pruning and additive Gaussian noise. When plotting perplexity under additive Gaussian noise, x-axis indicates the standard deviation of noise added to weights. Y-axis indicates the average perplexity over 20 runs with 95% confidence intervals. The second plot shows perplexity as we delete more weights based on magnitude, for the bilingual model at each epoch. X-axis indicates the probability of deleting sorted attention weight parameters. After only one epoch, the model shows higher sensitivity to weight perturbations. However, after eight epochs of training, it becomes more robust.

1 on task 1 (English Wikipedia dataset) for 16k more iterations. 2) We train a second model on a different English dataset, the LAMBADA dataset [16], for 16k more iterations. Figure 5 indicates the results of this experiment by plotting maximum singular values of the first attention layer. As the Figure shows, training model on a new dataset (task 2) results in a faster decay of the maximum singular value.

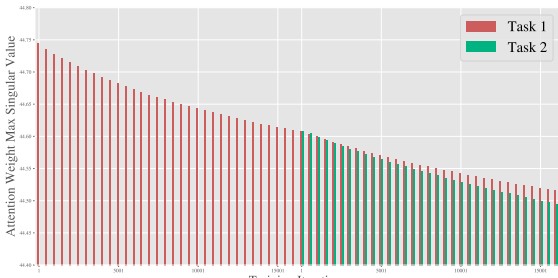

Figure 5: We show the effect of monolingual and bilingual training on the maximum singular value of attention weights. The red line shows the maximum singular value for a monolingual model trained on English Wikipedia for 32k iterations. The green line shows the maximum singular value if in the $16K$ iteration we switch to bilingual training. As shown, bilingual training leads to faster decay in the maximum singular value.

**Text Classification.** We conduct another set of experiments to observe the robustness of fine-tuned monolingual and bilingual GPT-2 models for text classification. In this section, we fine-tune both the monolingual and the bilingual GPT-2 models (previously trained) for downstream classification tasks using the GLUE benchmark [17, 18, 19, 20, 21, 22, 23, 24, 25, 26, 27, 28] to compare the robustness of models to weight perturbations. The two perturbation methods tested in this section are random weight deletion and random Gaussian noise added to attention weights. For each task, we fine-tune both models for ten epochs. When applying random pruning, the accuracy of each model is evaluated after deleting $p$ percent of model weights, $p$ ranging from $0\%$ to $45\%$. When perturbing model weights by adding noise, we try various Gaussian noise distributions with standard deviations ranging from 0 to 0.09. Experiment results can be found in the Appendix section.

**Random Pruning.** We compare the classification accuracy between the fine-tuned model from the monolingual pre-trained network and the fine-tuned model using the bilingual network. Each element in attention parameters is pruned with probability $p$, where $p$ ranges from .0 to .45. We evaluate the classification accuracy for the following GLUE tasks: CoLA, QQP, SST2, MRPC, QNLI, and RTE.

| Task | Fine-tuned using | |
| --- | --- | --- |
| | Monolingual ckpt | Bilingual ckpt |
| SST2 | 70567.875 | 60663.121 |
| QQP | 70608.195 | 60649.586 |
| MRPC | 70498.953 | 60590.769 |
| RTE | 70508.968 | 60590.765 |
| CoLA | 70519.781 | 60600.933 |

Table 1: We compute the sum of the squares of the weights of an attention layer for monolingual and bilingual models. The latter have smaller magnitudes, indicating that multitasking induces weight regularization.

| Pruning Probability | QQP | | SST2 | | COLA | | MRPC | | RTE | |
| --- | --- | --- | --- | --- | --- | --- | --- | --- | --- | --- |
| | m. | b. | m. | b. | m. | b. | m. | b. | m. | b. |
| 0.00 | 0.876 | 0.843 | 0.908 | 0.862 | 0.437 | 0.218 | 0.828 | 0.774 | 0.646 | 0.595 |
| 0.05 | 0.873 | 0.842 | 0.909 | 0.866 | 0.425 | 0.203 | 0.804 | 0.769 | 0.640 | 0.589 |
| 0.10 | 0.867 | 0.833 | 0.899 | 0.868 | 0.403 | 0.204 | 0.730 | 0.744 | 0.603 | 0.575 |
| 0.15 | 0.848 | 0.819 | 0.871 | 0.866 | 0.366 | 0.185 | 0.619 | 0.730 | 0.600 | 0.562 |
| 0.20 | 0.804 | 0.786 | 0.836 | 0.859 | 0.326 | 0.179 | 0.416 | 0.663 | 0.561 | 0.553 |
| 0.25 | 0.711 | 0.732 | 0.806 | 0.847 | 0.267 | 0.159 | 0.377 | 0.653 | 0.543 | 0.546 |
| 0.30 | 0.656 | 0.678 | 0.760 | 0.828 | 0.216 | 0.137 | 0.320 | 0.504 | 0.537 | 0.536 |
| 0.35 | 0.638 | 0.674 | 0.714 | 0.815 | 0.153 | 0.092 | 0.317 | 0.420 | 0.522 | 0.494 |
| 0.40 | 0.632 | 0.655 | 0.683 | 0.793 | 0.097 | 0.058 | 0.316 | 0.328 | 0.521 | 0.488 |
| 0.45 | 0.632 | 0.636 | 0.651 | 0.773 | 0.060 | 0.042 | 0.316 | 0.328 | 0.525 | 0.485 |

Table 2: Performance under a range of random pruning probabilities for various GLUE tasks. Columns labeled with "m" determine classification accuracy of monolingual models and columns labeled as "b" determine accuracy of bilingual. CoLA is evaluated using Matthew's Correlation and other tasks are evaluated by accuracy.

We expect the accuracy of both models to decay as we prune a more considerable number of parameters. The monolingual model shows a faster decay in almost all tasks. For some tasks such as SST2, QQP, and MRPC, we observe that the bilingual model starts with lower accuracy, and its performance exceeds the monolingual model as we prune $\approx 5\%$ to $\approx 25\%$ of parameters. A detailed set of results in Table 2 show models' average prediction accuracy on the GLUE benchmark.

**Random Noise.**  We also experiment with adding Gaussian noise to the weights. We vary the noise standard deviation from .0 to 0.09. We evaluate the classification accuracy for the same tasks. When no noise is added to model parameters, the monolingual model performs slightly better for tasks like QQP and SST2. As we increase the noise, the accuracies of both models drop with almost identical rates. However, both graphs illustrate a cross-over point after which the bilingual model outperforms the monolingual. The bilingual model achieves significantly higher accuracy in the MRPC task when the standard deviation is greater than $\approx 0.03$. For CoLA and RTE, the monolingual model maintains maintains higher performance regardless of the noise level. A detailed set of results in the Appendix section shows models' average prediction accuracy on the GLUE benchmark.

## 4   Related Work

**Cognitive Reserve and Bilingualism.** Our work is inspired by Cognitive Science and evidence of Cognitive Reserve in bilinguals. One implication of our theory is that multitasking leads to smaller weights on average. This could be related to studies performed in healthy older adults that indicate that despite overall less gray matter volume and poorer white matter integrity (i.e., poorer structural brain connectivity), older healthy bilinguals perform equally well or outperform monolinguals in several cognitive tasks [1, 2].

We would like to emphasize that our research is solely on *artificial networks* which have huge differences to biological neurons. No definite extrapolations should be made to Cognitive Neuroscience without further work. Nonetheless, we show that there is a simple mathematical abstraction that seems to align with the significantly more complex phenomena observed in bilingual cognitive reserve.

**Multitask Learning.** The most closely related work is by Mao et al. [29] which shows that multitask learning increases *adversarial* robustness. The intuition behind their proof is that, with task diversity, the gradient of the loss with respect to the wrong label is small as orthogonal tasks make gradients

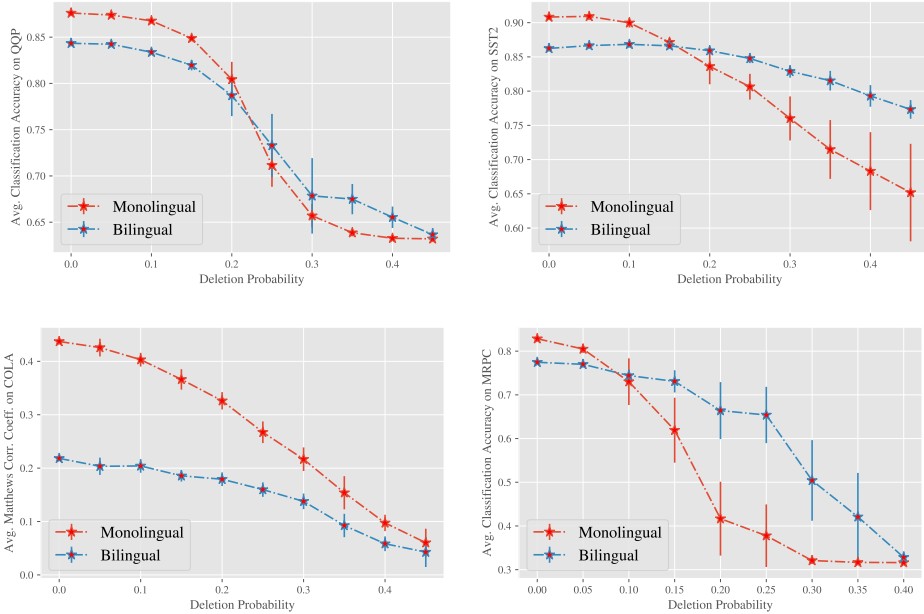

Figure 6: Performance comparison in GLUE tasks: QQP, SST2, CoLA, and MRPC under random erasures. QQP: Monolingual drops lower than the bilingual model after $\approx 25\%$ of the parameters are deleted. SST2: Monolingual drops with a faster rate, falling behind the bilingual after deleting $\approx 15\%$ of the parameters. CoLA: Both models reach $\approx 0$ MCC (random prediction) with $\approx 45\%$ of parameters pruned. MRPC: The accuracy of the monolingual degrades at a faster rate as pruning probability increases higher than $\approx 10\%$.

that cancel out. Wu et al. [30] establishes a connection between robustness to weight perturbations and adversarial attacks. Our work is related but different since it directly establishes a connection between structural robustness and multitasking and shows a cross-over in performance across various domains and tasks. Our theoretical analysis is also completely different compared to prior works. More information on multitask learning can be found in Mao et al. [29] and Ghamizi et al. [31].

Many studies on network compression and the Lottery Ticket Hypothesis are related to our Magnitude Pruning experiments. LeCun et al. [32], Han et al. [7] find that selectively pruned networks can be trained from randomly initialized weights to match the performance of the original network. Frankle and Carbin [33] introduces the hypothesis that randomly initialized neural networks contain a very sparse sub-network that, if initialized correctly, can achieve the accuracy of the original model. Chen et al. [34] studies this in continual learning and examines various pruning methods.

## 5 Conclusions

We demonstrated a connection between multitask learning and robustness to structural failures for artificial neural networks. For linear representation learning we obtained a characterization of robustness through the spectrum of the task matrix. We showed that robustness comes from diverse tasks which imply a bounded spectral norm for $C$. One limitation of our theoretical work is that we did not analyze learning algorithms but directly used the SVD solution. It would be interesting to see if gradient descent introduces further regularization or other effects, especially in the non-linear case.

Experimentally, we observed increased robustness for both linguistic and non-linguistic tasks. More complex settings like multi-lingual models, cross-language transfer and their interactions remain to be explored. Finally, it remains open if bilingualism and cognitive reserve in humans can indeed be connected to our framework. It would be fascinating if neuroimaging techniques can measure any form of anatomical or functional regularization that bilingualism could be creating in humans.

# 6  Acknowledgments

This research has been supported by NSF Grants CCF 1763702, AF 1901292, CNS 2148141, Tripods CCF 1934932, IFML CCF 2019844, the Texas Advanced Computing Center (TACC) and research gifts by Western Digital, WNCG IAP, UT Austin Machine Learning Lab (MLL), Cisco and the Archie Straiton Endowed Faculty Fellowship.

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
