# A   Proofs

**Theorem 2.5.** *Let $C \in \mathbb{R}^{d \times T}$ be a matrix with distinct singular values $\sigma_1 > \sigma_2 > ... > \sigma_T$. Let $W, \Gamma$ be the SVD solution of* (2). *Under the Additive noise model defined in 6,*

$$\overline{\text{MSE}}^{T,\sigma^2} = \overline{\text{MSE}}^{T,0} + \frac{\sum_{i=1}^{k} \sigma_i^2(C)}{T}\sigma^2 \ . \tag{12}$$

*Proof.*

$$\text{MSE}^{i,\sigma^2} = \mathbb{E}\left[\left(c_i^T x - \gamma_i^T W_c x\right)^2\right] = x^T c_i c_i^T x - 2c_i^T x \gamma_i^T \mathbb{E}[W_c]x + x^T \mathbb{E}[W_c^T \gamma_i \gamma_i^T W_c]x \tag{13}$$

$$= x^T c_i c_i^T x - 2c_i^T x \gamma_i^T W x + x^T \mathbb{E}[W_c^T \gamma_i \gamma_i^T W_c]x \tag{14}$$

$$= x^T c_i c_i^T x - 2c_i^T x \gamma_i^T W x + x^T \mathbb{E}[(W^T + N^T)\gamma_i \gamma_i^T (W + N)]x \tag{15}$$

$$= x^T c_i c_i^T x - 2c_i^T x \gamma_i^T W x + x^T W^T \gamma_i \gamma_i^T W x + x^T \mathbb{E}[N^T \gamma_i \gamma_i^T N]x \ . \tag{16}$$

Now, observe that:

$$\sum_{i=1}^{T} \text{MSE}^{i,\sigma^2} = \left(\sum_{i=1}^{T} x^T c_i c_i^T x - 2c_i^T x \gamma_i^T W x + x^T W^T \gamma_i \gamma_i^T W x\right) + x^T \mathbb{E}[N^T \gamma_i \gamma_i^T N]x \tag{17}$$

$$\overline{\text{MSE}}^{T,\sigma^2} = \overline{\text{MSE}}^{T,0} + \frac{x^T \mathbb{E}\left[N^T \left(\sum_{i=1}^{T} \gamma_i \gamma_i^T\right) N\right] x}{T} \ . \tag{18}$$

Observe that

$$\mathbb{E}\left[N^T \left(\sum_{i=1}^{T} \gamma_i \gamma_i^T\right) N\right] = \sigma^2 \text{tr}\left(\sum_{i=1}^{T} \gamma_i \gamma_i^T\right) I_d \ . \tag{19}$$

For any unit-norm $x$:

$$x^T \mathbb{E}\left[N^T \left(\sum_{i=1}^{T} \gamma_i \gamma_i^T\right) N\right] x = \sigma^2 \text{tr}\left(\sum_{i=1}^{T} \gamma_i \gamma_i^T\right) \ . \tag{20}$$

Now for the SVD solution, we know that $\Gamma = \Sigma_k V^T$ and $\{\gamma_i\}$ are the columns of $\Gamma$. Hence,

$$\sum_{i=1}^{T} \gamma_i \gamma_i^T = \Sigma_k^2 V^T V = \Sigma_k^2. \tag{21}$$

Then,

$$\overline{\text{MSE}}^{T,\sigma^2} = \overline{\text{MSE}}^{T,0} + \sigma^2 \frac{\sum_{i=1}^{k} \sigma_i(C)^2}{T} \ . \tag{22}$$

$\square$

We are going to use the following result due to Ledoux [35].

**Lemma A.1** (Ledoux [35]). *Let $C_T : \mathbb{R}^{d \times T}$ be a random matrix whose entries are i.i.d. Gaussian with variance $1/d$. Let $C_K$ be the random matrix that submatrix of $C$ that consists of the first $K$ columns of $C_T$. Then,*

$$\Pr\left[\sigma_{\max}(C_K) \geqslant 1 + \sqrt{K/d} + o(1) + \alpha\right] \leqslant \exp(-d\alpha^2/2) \tag{23}$$

*and*

$$\Pr\left[\sigma_{\min}(C_K) \geqslant 1 - \sqrt{K/d} + o(1) - \alpha\right] \leqslant \exp(-d\alpha^2/2) \ , \tag{24}$$

*where $o(1)$ is a small-term that tends to 0 as $d \to \infty$.*

**Theorem 2.7.** *Let $C \in \mathbb{R}^{d \times T}$ be a random matrix with Gaussian, i.i.d. entries of variance $1/d$ and $d = \Omega(T^3)$. Let $C_t, C_{t+1}$ be the matrices that are formed by selecting the first $t, (t+1)$ columns of $C$ respectively. Then, there is a noise level $\sigma_{\text{thres}}$ such that with probability $\geqslant 1 - \exp\left(-\Omega\left(\sqrt{d}\right)\right)$, the SVD solutions (see (4)) of (2) (for $C_t, C_{t+1}$ respectively), under the noise corruption model, satisfy:*

$$\overline{\text{MSE}}^{t+1,\sigma^2} < \overline{\text{MSE}}^{t,\sigma^2} , \tag{25}$$

$\forall \sigma \geqslant \sigma_{\text{thres}}$.

*Proof.* From Theorem 2.5, we have that:

$$\overline{\text{MSE}}^{t+1,\sigma^2} = \overline{\text{MSE}}^{t+1,0} + \frac{\sum_{i=1}^{k} \sigma_i^2(C_{t+1})}{t} \sigma^2, \qquad \overline{\text{MSE}}^{t,\sigma^2} = \overline{\text{MSE}}^{t,0} + \frac{\sum_{i=1}^{k} \sigma_i^2(C_t)}{t} \sigma^2 . \tag{26}$$

To prove the desired thing, we just need to show that $\overline{\text{MSE}}^{t+1,\sigma^2}$ has a smaller co-efficient for the term $\sigma^2$, because for large enough $\sigma$, eventually this term will dominate the sum. Hence, we need to show that:

$$\frac{\sum_{i=1}^{k} \sigma_i^2(C_t)}{t} \geqslant \frac{\sum_{i=1}^{k} \sigma_i^2(C_{t+1})}{t+1} . \tag{27}$$

Since $C_t$ is a submatrix of $C_T$, from the Eigenvalue Interlacing Theorem we know that $\sum_{i=1}^{k} \sigma_i^2(C_t) \leqslant \sum_{i=1}^{k} \sigma_i^2(C_{t+1})$. However, the difference of the two sums is upper-bounded. Using Lemma A.2, we get that:

$$\sum_{i=1}^{k} \sigma_i^2(C_{t+1}) = \sigma_1^2(C_{t+1}) + \sum_{i=2}^{k} \sigma_i^2(C_{t+1}) \tag{28}$$

$$\leqslant \sigma_1^2(C_{t+1}) + \sum_{i=1}^{k-1} \sigma_i^2(C_t) \tag{29}$$

$$= \sigma_1^2(C_{t+1}) - \sigma_k^2(C_t) + \sum_{i=1}^{k} \sigma_i^2(C_t). \tag{30}$$

It suffices to show that:

$$\frac{\sum_{i=1}^{k} \sigma_i^2(C_t)}{t} \geqslant \frac{\sigma_1^2(C_{t+1}) - \sigma_k^2(C_t) + \sum_{i=1}^{k} \sigma_i^2(C_t)}{t+1} \iff \tag{31}$$

$$\sum_{i=1}^{k} \sigma_i^2(C_t) \geqslant t \left(\sigma_1^2(C_{t+1}) - \sigma_k^2(C_t)\right). \tag{32}$$

Trivially, $\sum_{i=1}^{k} \sigma_i^2(C_t) \geqslant k\sigma_k^2(C_t)$. Hence, it is enough to show that:

$$\sigma_k^2(C_t) \geqslant \frac{t}{k} \left(\sigma_1^2(C_{t+1}) - \sigma_k^2(C_t)\right). \tag{33}$$

We will now bound the difference of the first and $k$-th singular values.

From Lemma A.1, we have that:

$$\Pr\left[\sigma_1(C_{t+1}) \geqslant 1 + o(1) + \sqrt{\frac{t+1}{d}} + \alpha\right] \leqslant \exp\left(-d\alpha^2/2\right) \tag{34}$$

and

$$\Pr\left[\sigma_k(C_t) \leqslant 1 + o(1) - \sqrt{\frac{t}{d}} - \alpha\right] \leqslant \exp(-d\alpha^2/2). \tag{35}$$

By union bound, with probability $\geqslant 1 - 2\exp(-d\alpha^2/2)$, we have that:

$$\sigma_1^2(C_{t+1}) - \sigma_k^2(C_t) \leqslant \left(1 + o(1) + \sqrt{\frac{t+1}{d}} + \alpha\right)^2 - \left(1 + o(1) - \sqrt{\frac{t}{d}} - \alpha\right)^2 \tag{36}$$

$$= 2(1 + o(1))\left(\sqrt{\frac{t+1}{d}} + \alpha\right)\left(\sqrt{\frac{t}{d}} + \alpha\right) + \left(\sqrt{\frac{t+1}{d}} + \alpha\right)^2 - \left(\sqrt{\frac{t}{d}} + \alpha\right)^2 \tag{37}$$

$$\leqslant 5\left(\sqrt{\frac{t+1}{d}} + \alpha\right)^2. \tag{38}$$

We choose $\alpha = \left(\frac{t+1}{d}\right)^{1/4}$. Since, $t < T < d$, we have that with probability $\geqslant 1 - \exp\left(-\Omega\left(\sqrt{d}\right)\right)$,

$$\sigma_1^2(C_{t+1}) - \sigma_k^2(C_t) \leqslant 20\sqrt{\frac{t+1}{d}}, \quad \sigma_k(C_t) \leqslant 1 + o(1) - 2\sqrt{\frac{t+1}{d}}. \tag{39}$$

Going back to Eq. 33, it suffices to show that:

$$1 + o(1) - 2\sqrt{\frac{t+1}{d}} \geqslant \frac{20t}{k}\sqrt{\frac{t+1}{d}} \iff 1 + o(1) \geqslant \sqrt{\frac{t+1}{d}}\left(\frac{20t}{k} + 2\right). \tag{40}$$

Since $t < T$, this is true for $d = \Omega(T^3)$.

$\square$

**Lemma A.2.** *Let $C$ be a matrix $\in \mathbb{R}^{d \times T}$ and $c_{T+1} \in \mathbb{R}^d$. Let also $\mathrm{C}_{\mathrm{new}} = \begin{bmatrix} C & c_{T+1} \end{bmatrix} \in \mathbb{R}^{d \times T+1}$. Denote with $\sigma_i(C)$ the $i$-th singular value of $C$, sorted from the largest to the smallest. Then,*

$$\sigma_{i+1}(\mathrm{C}_{\mathrm{new}}) \leqslant \sigma_i(C) \leqslant \sigma_i(\mathrm{C}_{\mathrm{new}}), \quad \forall i \in \{1, ..., T\} \tag{41}$$

*Proof.* We have that:

$$\mathrm{C}_{\mathrm{new}}{}^T\mathrm{C}_{\mathrm{new}} = \begin{bmatrix} C^T \\ c_{T+1}^T \end{bmatrix} \cdot \begin{bmatrix} C & c_{T+1} \end{bmatrix} = \begin{bmatrix} C^T C & C^T c_{T+1} \\ c_{T+1}^T C & c_{T+1}^T c_{T+1} \end{bmatrix}. \tag{42}$$

Observe that $\mathrm{C}_{\mathrm{new}}{}^T\mathrm{C}_{\mathrm{new}}$ is a symmetric matrix and $C^T C$ is a principal submatrix. Hence, from the Eigenvalue Interlacing Theorem, we have that:

$$\lambda_{i+1}(\mathrm{C}_{\mathrm{new}}{}^T\mathrm{C}_{\mathrm{new}}) \leqslant \lambda_i(C^T C) \leqslant \lambda_i(\mathrm{C}_{\mathrm{new}}{}^T\mathrm{C}_{\mathrm{new}}), \tag{43}$$

where $\lambda_i(A)$ is the $i$-th eigenvalue of $A$, sorted from the largest to the smallest. To finish the proof, we note that for any matrix $A$, $\sigma_i(A) = \sqrt{\lambda_i(A^T A)}$. $\square$

# B   Additional Results

In this section, we include additional results that further support the findings of the main paper.

**Robustness slope**   Recall Theorem 2.5 of the paper.

$$\underbrace{\overline{\text{MSE}}^{T,\sigma^2}}_{\text{Average MSE under noise}} = \underbrace{\overline{\text{MSE}}^{T,0}}_{\text{Average MSE without noise}} + \underbrace{\frac{\sum_{i=1}^{k} \sigma_i(C)^2}{T}}_{\text{Robustness slope}} \cdot \underbrace{\sigma^2}_{\text{Noise Variance}} . \tag{44}$$

This theoretical finding implies that the cross-over phenomenon that we observe in our experiments (at least for the linear case), stems from a lower Robustness Slope in the multitask models. Figure 3 shows that the MSE under noise is lower for models that are trained to do more tasks. In Figure 7 of this Appendix, we show that indeed this is due to a decrease in the robustness slope. Across three different datasets, MNIST, CIFAR10, NewsGroup20, we see that increasing the number of tasks leads to a decrease in the robustness slope. We note that this does not necessarily mean a monotonic decrease in the MSE under noise. Since the total dataset size and the parameter $k$ stay the same, increasing the number of tasks usually leads to increased noiseless MSE. However, under the presense of noise, our theory predicts (and our experiments confirm) that eventually the multitask model will reach superior performance.

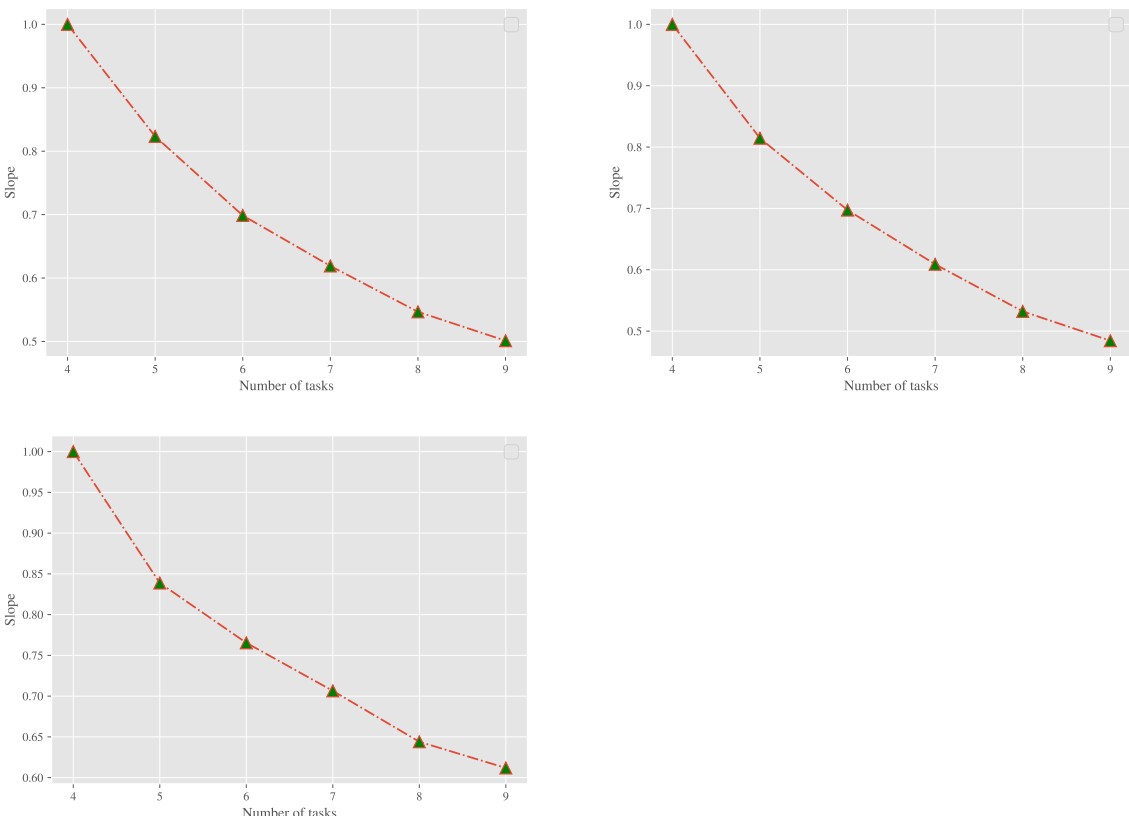

Figure 7: Slope as a function of the number of tasks for different datasets. (1, 1): MNIST, (1, 2): CIFAR10, (2, 1): NewsGroup20. As shown, adding more tasks decreases the robustness slope which leads to an increase in robustness (see Theorem 2.5).

**Experiments on other languages** For our experiments on multilingual generative models, we decided to use Greek and English because we were looking for a linguistic pair with different morphology, syntax and phonology. This is inspired by our theory on linear models that shows that diversity in the tasks (as we have for the Gaussian task vectors) leads to a sublinear increase in the sum of the top-k singular values of the task matrix and hence an increase in robustness. For completeness, we include here experiments on a different linguistic pair, English and Spanish. English and Spanish much closer linguistically and also share the Latin alphabet, so we expect bigger transfer and smaller robustness benefit in this linguistic pair.

We compare a monolingual English model (finetuned on English Wikipedia) with a bilingual, English and Spanish, model. The bilingual model is finetuned on a concatenation of English and Spanish Wikipedia. We make sure that the total dataset size is the same for the monolingual and the bilingual model, i.e. the bilingual model is exposed to half English data compared to the monolingual. This ensures that any benefits in terms of robustness are not coming from exposure to more data. We present results on random deletions in Figure 8 of the Appendix – this Figure is similar to Figure 1 of the paper, but instead of having English and Greek, we have English and Spanish. As shown in Figure 8, even though the two models are starting from roughly the same perplexity, the bilingual model exhibits higher structural robustness in the presence of weight deletions. This is consistent with the results we showed across this paper and indicates that the increased robustness is not specific to the choice of the linguistic pair.

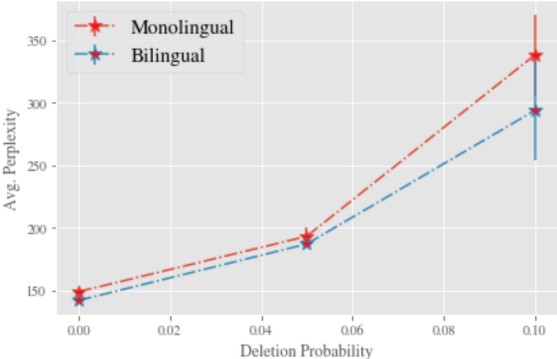

Figure 8: Performance of monolingual (English) and bilingual (English/Spanish) GPT-2 models with the same architecture and training dataset size. The x-axis indicates the probability of erasing an attention weight parameter (setting to it zero). The y-axis indicates the average perplexity over 20 runs. Models have a close initial accuracy. Perplexity increases (showing lower accuracy) as weight deletion probability is increased, though bilingual model perplexity rises at a slower rate.

Notice that the gap in the performance is smaller compared to the one presented in Figure 1. This is aligned with our theory for linear models that predicts that the benefits of multitasking for robustness are more evident for more diverse tasks. Since English and Spanish are linguistically closer, compared to English and Greek, our intuition is that the difference in robustness is going to be smaller and this is also confirmed by this experiment. An interesting future direction is to study this robustness benefit for multiple linguistic pairs or multi-lingual models. However, this study requires massive computational resources. Similarly, it would be interesting to study how the robustness gap in bilingual models scales as the datasets scale, but this also requires training multiple pairs of GPT models to comparable accuracy, and requires computational resources that were not available to us. We hope that future research is going to shed more light into these exciting directions.

**Experiments with different corruption mechanisms.** In the main paper, we primarily presented results with random deletions of neurons as our corruption model for the language modeling experiments. We include results for additive Gaussian noise for GPT-2 (monolingual and bilingual). We choose to present additional results with this noise model since it is the one analyzed by our theory. Table 3 summarizes how the performance of GPT-2 (monolingual and bilingual) changes when we add different amount of noise to the weights. We evaluate this performance on downstream tasks from the GLUE paper. Figure 9 visualizes the decrease of performance as the magnitude of the noise rises for different number of tasks. The results are similar with the results presented in the main paper for random deletions. In QQP, the monolingual model performs better without perturbations. Both models decay with a close rate. The monolingual model outperforms in SST-2 with no perturbations. Both models decay with a close rate. For CoLA, the monolingual model maintains a significantly better performance regardless of the noise level. Finally, for MRPC we see that although the bilingual model shows a weaker classification accuracy with no noise, it outperforms the monolingual model for noise levels higher than 0.035. These results complement Figure 4 of the main paper that shows robustness of GPT-2 to additive Gaussian noise for the task of language modeling.

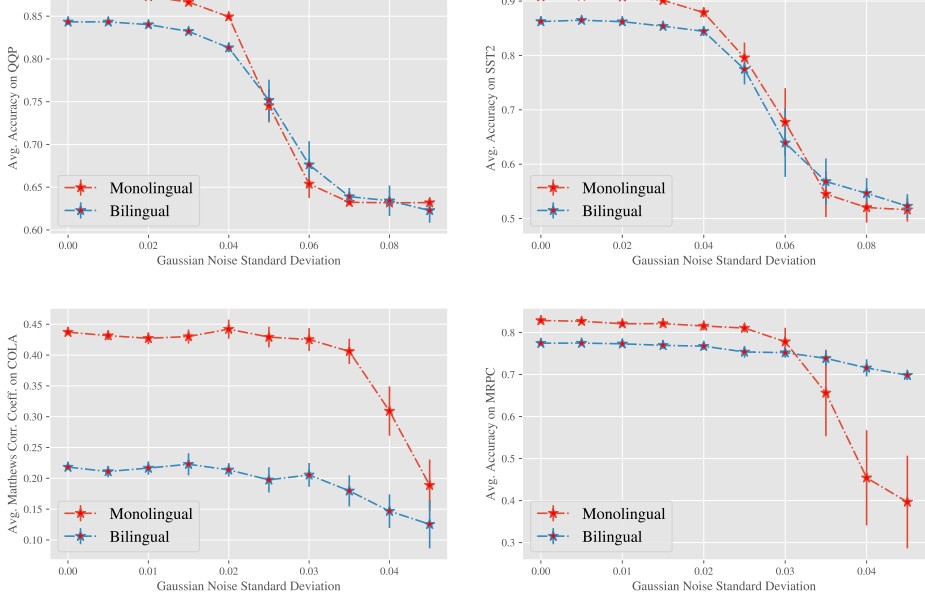

Figure 9: Performance comparison in GLUE tasks: QQP, SST2, CoLA, and MRPC under Gaussian noise. QQP: The monolingual model performs better without perturbations. Both models decay with a close rate. SST2: The monolingual model outperforms with no perturbations. Both models decay with a close rate. CoLA: The monolingual model maintains a significantly better performance regardless of the noise level. MRPC: Although the bilingual model shows a weaker classification accuracy with no noise, it outperforms the monolingual model for noise levels higher than 0.035.

| Gaussian std. | QQP | | SST2 | | COLA | | MRPC | | RTE | |
| --- | --- | --- | --- | --- | --- | --- | --- | --- | --- | --- |
| | m. | b. | m. | b. | m. | b. | m. | b. | m. | b. |
| 0.00 | 0.876 | 0.843 | 0.908 | 0.862 | 0.437 | 0.218 | 0.828 | 0.774 | 0.646 | 0.595 |
| 0.01 | 0.875 | 0.843 | 0.909 | 0.864 | 0.432 | 0.216 | 0.827 | 0.772 | 0.639 | 0.597 |
| 0.02 | 0.873 | 0.840 | 0.907 | 0.862 | 0.444 | 0.213 | 0.816 | 0.760 | 0.641 | 0.591 |
| 0.03 | 0.866 | 0.832 | 0.901 | 0.853 | 0.440 | 0.205 | 0.776 | 0.749 | 0.643 | 0.590 |
| 0.04 | 0.849 | 0.813 | 0.878 | 0.844 | 0.316 | 0.146 | 0.494 | **0.711** | 0.634 | 0.589 |
| 0.05 | 0.745 | **0.751** | 0.795 | 0.774 | 0.088 | 0.074 | 0.326 | **0.673** | 0.622 | 0.577 |
| 0.06 | 0.653 | **0.676** | 0.677 | 0.639 | -0.002 | 0.004 | 0.316 | **0.610** | 0.602 | 0.563 |
| 0.07 | 0.632 | **0.638** | 0.545 | 0.568 | 0.019 | 0.002 | 0.316 | **0.577** | 0.585 | 0.562 |
| 0.08 | 0.631 | **0.634** | 0.520 | **0.546** | -0.006 | -0.006 | 0.316 | **0.465** | 0.539 | 0.546 |
| 0.09 | 0.631 | 0.622 | 0.516 | **0.522** | -0.014 | 0.002 | 0.316 | **0.451** | 0.536 | 0.528 |

Table 3: Performance on GLUE when adding Gaussian noise. Columns labeled with "m" determine classification accuracy of monolingual models and columns labeled as "b" correspond to bilingual models. CoLA is evaluated using Matthew's Correlation and other tasks are evaluated by accuracy.

> **[the company produces]** video games, television programs, and online services. the company is headquartered in new york city and is the world's second largest entertainment company in terms of revenue, after comcast. disney was founded on october 16, 1923, by brothers walt disney and roy o'brien, jr.
>
> **[η εταιρεία παράγει]** και διανέμεται στο χρηματιστήριο αθηνών, το οποίο παρέχει υπηρεσίες για τη διαχείριση των υπηρεσιών της εταιρείας. η εταιρεία είναι επίσης υπεύθυνη για την προώθηση του διαδικτύου σε συνεργασία με άλλες εταιρείες που δραστηριοποιούνται στην ευρωπαϊκή ραδιοτηλεοπτική ένωση

Table 4: Sample text generated by the bilingual GPT-2 model. Text in the brackets is the starting prompt provided for model.

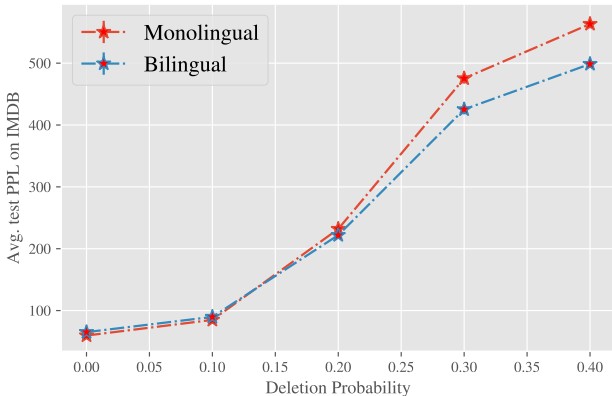

Figure 10: Performance of monolingual and bilingual GPT-2 models with the same architecture and training dataset size. Models are trained using the truncation length 1024. We show the performance as we randomly erase weights. The plot indicates the average perplexity over 20 runs with 95% confidence intervals. This plot indicates that the monolingual model declines faster and performs worse in the highly damaged regime. The bilingual GPT-2 model is more robust to neuron weight erasures.

## C    Training Details

Using the GPT-2 small model as our baseline, we fine-tuned a monolingual (English) model and a bilingual (English and Greek) model on Wikipedia text data. With set the vocabulary size to 50257 tokens. In both training processes, we set the initial learning rate to 3e-4 and configured a cosine learning rate scheduler with 150 warmup steps, setting AdamW optimizer weight decay to 0.01. We trained each model for eight epochs, using 4 NVIDIA Quadro RTX 5000 GPUs. Training took approximately 10 hours per epoch.

To fine-tune another bilingual model on English and Spanish data, we fine-tuned a monolingual model and a bilingual model on Wikipedia text data. With a vocabulary size of 50257 tokens, the monolingual model was fine-tuned on 800,000 English articles. The bilingual model was fined tuned on a mix of 400,000 Spanish and 400,000 English articles, using the same vocabulary size of 50257. Like the previous experiment, we set the initial learning rate to 3e-4 and configured a cosine learning rate scheduler with 150 warmup steps, setting AdamW optimizer weight decay to 0.01.

We further tuned bilingual and monolingual models for the text classification experiments using GLUE datasets. For these experiments, we used the AdamW optimizer with a learning rate of 2e-5, and epsilon at 1e-8. We used a linear scheduler with no warmup steps and trained models for more than ten epochs.

For the experiments on the linear representation layer, we used Adam optimizer with weight decay $1e - 4$. We trained all the models with a batch size of $128$ and to a maximum of 50 epochs. To emulate multiple tasks, we selected different subsets of the classes. We experiments with having class overlaps (e.g. for MNIST one task might have been predicting 0 vs 1 and some other task predicting 1 vs 2) and without class overlaps (e.g. predicting 0 vs 1 and 2 vs 3). We noticed bigger robustness benefits when there was no class overlap something that is consistent with our theoretical analysis that implies that diversity in the tasks is needed. In terms of corruptions, we also did preliminary experiments on random deletions and we saw similar results. The interested reader might use the released code to perform other types of weights corruptions and see how this affects robustness trends. For all our experiments, we fix the representation dimension to $k = 4$, which is also why we show the robustness slope from $k = 4$ onwards on Figure 7. Training time of the linear experiments depends on the dataset size: it took us roughly 1 hour on CIFAR-10 and 3 hours on NewsGroup20.

While preparing the camera-ready for this paper we noticed a parameter mismatch between the training parameters of the monolingual and the bilingual model affecting some of our experiments. Specifically, the bilingual training dataset was tokenized with sentences truncated at length 128 while the monolingual dataset was tokenized using truncation length 1024. To check if this hyperparameter

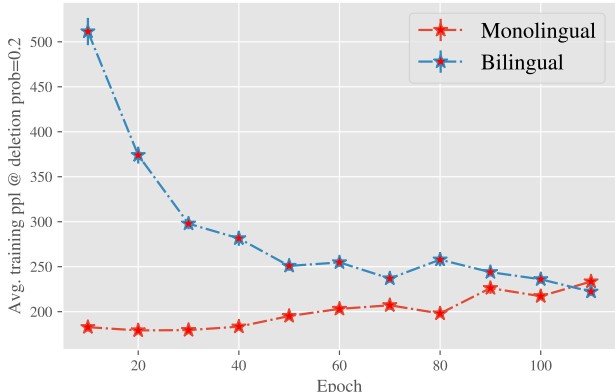

Figure 11: The plot shows performance of the monolingual and bilingual models during training. We plot performance at 20 percent random weight deletions to observe robustness behavior as models are trained. Notice that the bilingual model starts with very poor performance because the tokenizer has been randomized due to the recycling process. As training progresses, the bilingual model outperforms the monolingual model and becomes more robust.

mismatch causes the bilingual robustness benefit, we re-trained the bilingual model using sequence truncation length 1024. This experiment is shown in Figure 10 and shows similar behaviour as before.

We performed an additional experiment to study how robustness changes during training. In Figure 11 we plot the performance of the monolingual and bilingual models as we train, but plotting performance at 20 percent random deletions, i.e. measuring also robustness. We show that as training progresses, the bilingual model becomes more robust compared to the monolingual one.

The training datasets for the bilingual and monolingual models have the same size. The bilingual dataset is a concatenation of half Greek (from Greek Wikipedia) and half English text (from the cc_news dataset [36]). Similar to previous experiments, we use the Language Model Recycling Technique [13] to pre-trained GPT-2 (small)[6].

## D    Things that did not work

In the early stages of the project, we attempted to train a bilingual model from scratch, instead of using the recycling technique [13]. The dataset for the Greek model consists of roughly 2GB of text from Wikipedia. With such limited amount of data, we found it impossible to train a bilingual model that reaches a reasonable perplexity. Note that GPT-2 was trained on $\approx 40$GB of text, i.e. on a $\approx 20\times$ bigger dataset. We found that the recycling technique [13] enables learning with much smaller datasets (on top of the computational benefits it offers).

## E    Limitations and Ethical Considerations

**Limitations**    Even though the models we train can produce text of reasonable quality (e.g. see Table 4), they do not perform on par with state-of-the-art generative networks. There are many reasons for that, e.g. we do not have the computational resources to train bigger networks and the dataset size is small. Nevertheless, the goal of this paper is not to advance the state-of-the-art in text-generation but to shed light on how multitasking is related to robustness.

The motivation of this paper was a theory from Cognitive Science about increased robustness in bilingual speakers. We see that bilingual artificial networks are also more robust compared to monolingual models trained under the same setting. However, it is important to state that no definite extrapolations should be made to Cognitive Neuroscience without significantly much work. Our models of corruptions happening to the neural network's weights are chosen primarily for simplicity in the implementation and in the analysis. There is no evidence that brain pathologies have any resemblance to the models of corruption analyzed in this work for artificial neural networks.

Finally, our theory did not analyze the learning dynamics for approximating the task vectors. Instead, it used the SVD solution. Different choices of learning algorithms might lead to different behaviors regarding robustness. For example, for the linear case we showed that multitasking creates weight regularization. Higher explicit weight regularization (e.g. with high weight decay) might help the single task model decrease the robustness gap with the multitask networks. It would also be interesting to explore how the theory can be generalized to the non-linear case.

**Ethical Considerations**    As part of this work, we are releasing pre-trained bilingual models. Big language models can be misused in many different ways including spreading of fake news, generation of toxic speech, etc. We encourage the readers to refer to Bender et al. [37], Brown et al. [38] for an extended discuss of the risks of releasing powerful language models. In our case, the released models are not nearly as big or powerful as state-of-the-art networks such as GPT-3. For all our experiments, we are using the small version of GPT-2 and the main objective is to see how learning multiple languages affects robustness to weight corruptions. Additionally, we are not training these models from scratch, but we are using the recycling technique proposed in de Vries and Nissim [13], hence the environmental cost of the training is much smaller.

# F    Code and License

We open-source our code and pre-trained models to encourage more related research: https://github.com/giannisdaras/multilingual_robustness. The code us released under the GNU GEN-ERAL PUBLIC LICENCE. The interested reader should also refer to the licenses of pre-existing software we use. Please look at the `requirements.txt` file of our code to find all our dependencies.

The code for the training of the bilingual models is written in PyTorch [39] and it is based on the implementation of GPT-2 found in the `transformers` [40] library. The code for the linear experiments is written in JAX [41].

We expect that the release of bilingual and monolingual models trained on identical conditions will motivate further research in this area by cognitive scientists doing computational research. The main motivation for this paper was a theory from Cognitive Science regarding increased Cognitive Reserve in bilingual people. We expect that there could be many more interesting directions in Cognitive Science that can be studied from a computational perspective and we hope that the release of bilingual models will contribute towards this goal.