# OpenReview forum: "Multitasking Models are Robust to Structural Failure: A Neural Model for Bilingual Cognitive Reserve"
_NeurIPS.cc/2022/Conference — NeurIPS 2022 Accept_

### Official Review · Reviewer_1N4w · 2022-07-05

**Rating:** 7
**Confidence:** 3
**Soundness:** 3 good
**Presentation:** 3 good
**Contribution:** 3 good

**Summary:**

This paper argues that multi-task training functions as a regularizer in neural network optimization, yielding models which are more robust to different types of structural failures (perturbations or dropouts of weights). They demonstrate this formally by analyzing a representation learning task in which a model learns to predict multiple downstream task representations, and show a connection to L2-regularization under assumptions of the structural failure / noise process and orthogonality of task target representations. They next test the idea empirically in a broad set of experiments in multiple modalities (vision and language), and show that multi-task learning leads to slower performance decay (relative to single/fewer tasks) as the degree of various weight perturbations increases.

**Questions:**

1. The theory section of the paper reaches for an interesting connection between multitask training and L2 regularization, and this is later channeled in the weight norm table (Table 1) of the experimental section. But why not make this connection stronger in the experimental section, by using L2 regularization as a baseline / comparison case? I would expect to find that strong L2 leads to better performance, but that L2 models would still decay more quickly than multi-task trained models. The remaining gap in "cognitive reserve" between L2 and multi-task models would be explanatory work for future papers :)
2. The tasks in figure 3 don't seem to show a crossover point like the later language tasks (in which models trained on more tasks perform poorly in low-noise regimes, but perform better than fewer-task models in high-noise regimes). Why not?
3. Minor comments
	1. Figure 4 (right) makes it hard to see the intended trend (change in performance as a function of epochs). Can we see a slice of the current x-axis with epoch on the x-axis? Or plot pruning probability as color and epoch as x-axis?
	2. Figure 5 caption doesn't match text description. Is Task 2 bilingual training (according to caption) or training on English LAMBADA (according to main text description)?
	3. Not sure, but it might be better to plot perplexity on a log scale in figure 1, 4 so that we can better see the differences between the two models.

**Limitations:**

Yes.

**Strengths And Weaknesses:**

I think the paper presents an interesting idea, and the theory and experiments do a good job at backing up the claims as far as I can see.

- The paper combines a clear theoretical presentation with several stages of theoretical experiments. But the theory addresses the properties of an ideal (SVD) solution rather than investigating actual optimization methods. This gap would be less problematic if the authors included L2 regularization as a baseline / comparison case in the experimental section (see Question 1 below).
- The intuitive presentation is quite clear and worth the space in the paper, I think.
- The link to bilingualism is interesting but I think not entirely necessary. While the path leading the authors to their discovery is interesting, there doesn't seem to be a necessary conceptual link in either direction: the theory or experiments aren't necessarily about bilingualism (certainly not -- there are experiments on vision tasks!), and the authors themselves say that these results shouldn't be taken to imply anything strong for the cognitive neuroscience of language/bilingualism.

---

> ### Author Response · Authors · 2022-08-03
> **Reply to Reviewer 1N4w**
>
> Thank you for your positive and constructive feedback! Please find the answers to your questions below.
>
> > But the theory addresses the properties of an ideal (SVD) solution rather than investigating actual optimization methods. This gap would be less problematic if the authors included L2 regularization as a baseline / comparison case in the experimental section (see Question 1 below).
>
> This is a fair point. We used the SVD solution because it is the cleanest for theoretical analysis. We will add one more experiment to strengthen our paper as suggested if the reviewers agree: We compare the SVD solution and the solution we get by learning with gradient descent in rank k space + L2 regularization. For both SVD and L2 regularized GD solutions, we observe what happens to the MSE as we increase the noise, the amount of regularization, and the number of tasks. The results are shown in the following anonymous links and will be included in the camera-ready version of our paper:
> SVD solution: https://imgur.com/a/Pm1ru1a
> GD + L2 solution: https://imgur.com/a/Sgcl9WY
>
> For the SVD solution, the noiseless accuracy for number of tasks=4 is ~70%. For the GD + L2 the accuracy in the noiseless case is ~63% for number of tasks=4.
>
> There are a couple of things we observe: First, L2 regularization indeed improves robustness to weight shifts as expected. This is consistent with our theoretical predictions. In our theoretical results, we show that for linear representation learning, the robustness to noise addition to the weights depends on the sum of the first k-singular values of the task matrix C (see Theorem 2.5).
>
>
> (Side note, Simple intuition for our theorem): The matrix C has T columns, one for each task. Assume for simplicity that each column of C has unit norm. As more tasks are added, C gets more columns. Our first result is that robustness depends on the sum of the top k singular values of C and the representation dimension k, stays constant as T grows. We prove that if this sum of the top k singular values grows sublinearly in T, more tasks yield more robustness. Now, if all the task vectors (the unit columns of C) are perfectly aligned, the top singular value of C scales linearly in T (the strongest direction gets more and more energy as tasks are added) and hence we get no robustness or regularization from multitasking. On the other extreme, if each new task vector is orthogonal, adding new vectors in C does not increase the top-k singular value sum and robustness improves like 1 over T. This hopefully makes it clear why task diversity is needed for robustness, and random tasks are indeed near orthogonal. Our results on real tasks show that real task vectors are also sufficiently diverse. If the reviewers like this example we can add it to the paper.
>
> From this intuition, we hope it is clear that multitasking creates regularization ONLY when the task vectors are near orthogonal, but certainly not if they are near co-linear. On the contrary, explicitly adding L2 regularization is independent of the structure of the task vectors.
>
> The fact that L2 regularization improves robustness is very intuitive if one thinks about what happens to the limit: as we increase the weight of L2 regularization to infinity in the loss function, we will get the zero solution, which is robust to weight corruptions (e.g. performance does not decay as you increase noise), but useless for any predictions (starting accuracy is equal to random chance). The latter point leads us to the following observation: L2 regularization is not enough to achieve good robustness with reasonable accuracy. In the plot for GD + L2, we see that when the number of tasks is low (k=4 and number of tasks=4), we don’t have good robustness. The starting accuracy (without noise) for this plot is 63%, while the accuracy for the SVD (and the non-regularized GD) is ~70%. Contrary to that, increasing the number of tasks leads to modest accuracy drops but dramatically increases the robustness. For number of tasks 7, accuracy without noise drops from 70% to 68% in the noiseless case, but robustness increases just by adding new tasks, without any explicit L2 regularization (as shown in the SVD plot and as predicted by our theory).
>
> We finally want to note that multitasking and L2 regularization can be combined for controlling in the best way how good the initial performance is and how slow is the decay in the performance as we increase the corruption. For us, the most interesting thing is that multitasking offers this robustness implicitly and depends on the tasks.

---

> > ### Author Response · Authors · 2022-08-03
> > **Reply part 2**
> >
> >
> > > I would expect to find that strong L2 leads to better performance, but that L2 models would still decay more quickly than multi-task trained models. The remaining gap in "cognitive reserve" between L2 and multi-task models would be explanatory work for future papers :)
> >
> > We hope we made the difference clear from the previous analysis. Cognitive reserve is only due to task diversity and can be combined with explicit L2 if needed.
> >
> >
> > > The tasks in figure 3 don't seem to show a crossover point like the later language tasks (in which models trained on more tasks perform poorly in low-noise regimes, but perform better than fewer-task models in high-noise regimes). Why not?
> >
> >
> > We thank the reviewer for this reasonable question. There is a cross-over, but in the way we visualized it, it can’t be seen clearly in the plot. Specifically, in the plot, all models appear to have the same performance in the noiseless case. By inspecting the actual numbers that correspond to the plot points, we confirmed that for noise=0.0, the higher the number of tasks, the higher is the MSE. The plot shows that for higher noise levels, we get lower MSE when the number of tasks is high, so cross-over is happening. We will update the Figure to make the cross-over visible.
> >
> > > Figure 4 (right) makes it hard to see the intended trend (change in performance as a function of epochs). Can we see a slice of the current x-axis with epoch on the x-axis? Or plot pruning probability as color and epoch as x-axis?
> >
> > Thanks a lot for the suggestion. Please see the updated figure here: https://imgur.com/a/sA6R8De
> > If the Reviewer likes it, we can update our Figure in the next revision of our work.
> >
> > > Figure 5 caption doesn’t match the text description. Is Task 2 bilingual training (according to caption) or training on English LAMBADA (according to main text description)?
> >
> > Thanks for catching this! Figure 5 compares the singular value trends while training under 2 different tasks. Its caption should be “The green line shows the maximum singular value if in the 16K iteration we switch to the LAMBADA dataset.”
> >
> > > Not sure, but it might be better to plot perplexity on a log scale in figure 1, 4 so that we can better see the differences between the two models.
> >
> > We thank the reviewer for the suggestion – we will try to update accordingly in the revision of our work.

---

### Official Review · Reviewer_QnuC · 2022-07-10

**Rating:** 5
**Confidence:** 4
**Soundness:** 2 fair
**Presentation:** 2 fair
**Contribution:** 3 good

**Summary:**

The paper is inspired by cognitive research studies where bilingualism increases brain robustness and demonstrates whether artificial neural networks are more robust when trained on multiple languages or tasks. The theoretical justification was provided by analyzing linear multitask representation learning across three types of corruption: Additive Gaussian noise to the weights, random weight pruning, and magnitude-based weight pruning. The experimental studies on both vision tasks using CNNs, and language tasks using monolingual and bilingual GPT-2 models establish that multitasking leads to higher robustness for diverse task vectors.

**Questions:**

* What is the difference in the brain's language network for monolingual vs. bilingual people?
* What are linguistic properties missing in mono vs. bilingual people with aging or neural degeneration?
* What are linguistic properties active in mono vs. bilingual people despite aging or neural degeneration?
* Similarly, all the above questions need to be addressed for Neural networks as well.
* Why do authors add noise to the weights of language models? Can we infer anything from the human brain? A clear justification is needed

Minor Comments:
Typos:
* Conclusion: For linear representation learning we obtained a characterization (, is missing before we)
* No citations for GPT-2 model when first time was introduced in the introduction.
* Missing citations for datasets and many acronyms



**Ethics Review Area:**

["I don’t know"]

**Limitations:**

* The authors presented the problem clearly; however, the scope and applications of the problem are missing.
* Authors could check the weaknesses section and add more limitations if they have not been addressed in this paper.

**Strengths And Weaknesses:**

Strengths:
* Interpretation of both monolingual and bilingual language models and how these models connect to structural robustness is interesting.
* The paper formed a theoretical basis based on earlier cognitive bilingual studies and concluded that robustness comes from diverse tasks.
* Another strength of the paper is its vast number of experiments.


Weaknesses:

* The authors, inspired by cognitive science research, indicate that bilingualism increases brain robustness by reducing the rate of cognitive decline due to aging.
* However, the current work lacks a discussion on some known results from neuroscience?
* In particular, I would have liked to see a discussion on how the brain learns the language structure of individuals who speak a single language vs. bilingual and neural networks.
* It is unclear which linguistic properties are missing because of adding noise to neural network models?
* Can the authors associate the linear mapping between these single or multi-task network models with mono or bilingual brain datasets?
* Why do authors add noise to the weights of language models? Can we infer anything from the human brain? A clear justification is needed.

---

> ### Author Response · Authors · 2022-08-02
> **Reply to Reviewer QnuC**
>
>
> Thank you for your input. Answers to your comments:
>
> >In particular, I would have liked to see a discussion on how the brain learns the language structure of individuals who speak a single language vs. bilingual and neural networks. Questions: What is the difference in the brain's language network for monolingual vs. bilingual people?
> ​​
> As we explain in our paper, we are only inspired by human brain networks and significantly more research is needed before we can make any specific claims about humans. Still, we show that multitasking training in standard artificial neural networks can be a simple model for this phenomenon. Our work can have useful implications for artificial neural networks, see our answer to the applications question. Still, we will expand the discussion about bilingualism and brain differences in the paper, based on these replies, if this is satisfactory to the reviewers.
> Additional discussion we plan to add:
>
> Brain differences in monolingual vs bilingual people have been extensively studied. A “core language” network seems to be engaged in both monolinguals and bilinguals, which consists of the Broca’s area, the pre-supplementary motor area (pre-SMA) and the premotor area (Li et al., 2021). However, there are structural and functional differences on executive functioning of those universal language networks (Wong et al., 2016). Most structural brain differences between bilinguals and monolinguals have been observed in frontoparietal brain areas that are associated with executive functioning and cognitive control. For example, higher grey matter density in inferior parietal cortex has been found in bilinguals compared to monolinguals (this difference was sensitive to L2 age of acquisition and proficiency; Mechelli et al., 2004). Regarding brain function, studies have found increased activation in dorsolateral prefrontal cortex (DLPFC), which is linked to working memory, during linguistic tasks in bilinguals compared to monolinguals (e.g., Kovelman et al., 2008; Jasinka & Petitio, 2014).
>
> References we will include:
>
> Li, Q., Pasquini, L., Del Ferraro, G., Gene, M., Peck, K., Makse, H., Holodny, A. (2021). Monolingual and bilingual language networks in healthy subjects using functional MRI and graph theory. Scientific Reports, 11:10568.
>
> Wong, B., Yin, B., & O’Brien, B. 2016. Neurolinguistics: Structure, function and connectivity of the bilingual brain. BioMed Research International, 7069274.
>
> Mechelli, A., & Crinion, J., Noppeney, U., O’Doherty, J., Ashburner, J., Frackowiak, R., & Price, c. (2004). Neurolinguistics: Structural plasticity in the bilingual brain. Nature, 431(7010):757.
>
> Kovelman, I., Baker, S., & Petitto, A. (2008). Age of first bilingual language exposure as a new window into bilingual reading development. Bilingualism: Language and Cognition, 11, 203-223.
>
> Jasinska, K., & Petitto, A. (2014). Development of neural systems for reading in the monolingual and bilingual brain: New insights from functional near infrared spectroscopy neuroimaging. Developmental Neuropsychology, 39, 421-439.

---

> > ### Author Response · Authors · 2022-08-02
> > **reply part 2**
> >
> > On your next questions:
> > >What are linguistic properties missing in mono vs. bilingual people with aging or neural degeneration?
> > >What are linguistic properties active in mono vs. bilingual people despite aging or neural degeneration?
> >
> > Apart from the differences presented above on executive functions and cognitive control, bilingualism has been suggested as a protective factor against the onset of symptoms of dementia and it can delay those symptoms by 4-5 years (e.g., Bialystok et al., 2007). Behavioral data from large-scale studies indicate that the number of languages someone speaks can significantly predict verbal ability, measured by semantic judgment tasks (Ihle et al., 2016) as well as performance on cognitive screening tasks (e.g., Katzman cognitive test, Mini-Mental State Exam) that include tasks such as time orientation, immediate or delayed memory, naming, repetition, reading etc., with multilinguals performing better in such tests than monolinguals (e.g., Kave et al., 2008) .
> >
> > Related references we will add:
> >
> > Bialystok, E., Craik, F., & Freedman, M. (2007). Bilingualism as a protection against the onset of symptoms of dementia. Neuropsychologia, 45, 459-464.
> >
> > Ihle, A., Oris, M., Fagot, D., & Kliegel, M. (2016). The relation of the number of languages spoken to performance in different cognitive abilities in old age. Journal of Clinical and Experimental Psychology, 38:10, 1103-1114.
> >
> > Kave, G., Eyal, N., Shorek, A., & Cohen-Mansfield, J. (2008). Multilingualism and cognitive state in the oldest old. Psychology and Aging, 23, 70-78.
> >
> >
> >
> > >Similarly, all the above questions need to be addressed for Neural networks as well.
> >
> > For artificial neural networks our paper is analyzing these questions in numerous settings and under various modes of corruption as we explain below.
> >
> >
> > >Why do authors add noise to the weights of language models? Can we infer anything from the human brain? A clear justification is needed
> >
> > Our scientific hypothesis in the paper is that multilingual and multitask training creates structural robustness in artificial neural networks. For this reason, we explore various ways of training with multiple tasks and various ways of measuring structural robustness. Structural robustness means that the network shows graceful performance degradation as the weights are corrupted or deleted. For this reason we tried 3 corruption processes: Adding noise, deleting random weights and magnitude-based weight pruning, i.e. deleting the smallest weights first (a method used for network sparsification). These methods are not supposed to be mimicking human brain damage,  they are only simple  and natural corruption processes. The fact that we observe robustness in all three is strong evidence of an interesting phenomenon.
> >
> > The authors presented the problem clearly; however, the scope and applications of the problem are missing.
> >
> > This is a good question: Please see the general reply on applications.

---

### Official Review · Reviewer_WfPh · 2022-07-10

**Rating:** 7
**Confidence:** 2
**Soundness:** 3 good
**Presentation:** 3 good
**Contribution:** 3 good

**Summary:**

The paper introduces the hypothesis that neural networks are more resilient to structural noise when they have been trained to solve multiple tasks, rather than a single one. The hypothesis finds inspiration in a linguistic phenomenon under which cognitive functions of multilingual speakers are more resilient to dementia and cognitive decline. The paper proposes a formal explanation for this phenomenon based on an analysis of a linear case in which the task is that of approximating a number $T$ of target vectors $c_i$ using linear combinations of an orthonormal weight matrix $W^T$. In this case, they show that increasing the number of approximated tasks effectively decreases both the expected MSE of the solutions and the Frobenius norm of the linear coefficients (thus inducing L2 regularization on them), thus making a connection between structural robustness and L2 regularization. Then, the authors experiment both with linear representation layers on image recognition datasets and on language modelling, observing that models trained on more than 1 task tend to be more robust to different kinds of perturbations.

**Questions:**

1. It is not clear to me whether the experiments were adequately controlled. Were the "monotask" and "multitask" models trained for a comparable number of iterations on a comparable number of data points?
2. $T$ as in number of tasks and $T$ in $W^T$ are completely different $T$s, right? The second stands for transpose? If so, this is a tad confusing. I would strongly recommend using a different symbol for the transpose (see https://tex.stackexchange.com/questions/30619/what-is-the-best-symbol-for-vector-matrix-transpose) or using a different letter for the number of tasks.
3. The image recognition experiments are performed on the linear case. What do you think would change if using a CE loss or corrupting a non-linear network as in the LM experiments?

**Limitations:**

The authors have acknowledged that the studied networks are not state-of-the-art, but that is also not relevant to the question they want to study. They also make clear that beyond the loose inspiration on the cognitive phenomenon of "Bilingual Cognitive Reserve", no extrapolations should be made without significant work. To stay true to that acknowledged limitation, the authors could revise the title fragment "A Neural Model for Bilingual Cognitive Reserve", which for the uncareful reader could warrant such extrapolations.

**Strengths And Weaknesses:**

**Strengths**
1. This paper presents an intriguing hypothesis on the properties of optimal solutions to multi-task learning setups.
2. It presents an insightful theoretical analysis of the linear case that could possibly be extended in the future to account for more realistic scenarios.
3. Backs the theoretical claim with multiple experiments on different datasets.
4. The paper is reasonably clear, but there are several points that would need clarification (see Weaknesses below).

**Weaknesses**
1. Some parts of the experimental paradigm look a bit arbitrary. For instance, the authors use IMDB as an evaluation set. Why that is not clear. Do the results replicate, say, on validation sets of English Wikipedia? A second experiment uses the LAMBADA dataset as a secondary task. LAMBADA is *not* a language modeling dataset. It's just an evaluation dataset containing only a few selected passages, so it is not clear why it is a good choice for these experiments. It is also not clear what is the connection between the attention weights and the predicted effect on L2 regularization.
2. There seem to be missing references that come up on a quick google search, such as this one: https://arxiv.org/pdf/2004.11072.pdf
3. The paper introduces experiments, whose results are only presented in the appendix. This is not appropriate. Either these experiments should be only briefly mentioned in the main body, or the full results should be discussed in it. Only introducing them seems like a way of working around the page limit.
4. There seems to be a strong jump between the predictions in the linear case to the non-linear one. The latter ones are only explored in the Transformer LM experiments. It would have been good to exploit the simpler image recognition experiments to also explore whether the results followed the predictions in non-linear networks trained with CE.

---

> ### Author Response · Authors · 2022-08-02
> **Reply to Reviewer WfPh**
>
>
> Thank you for your positive and constructive input. Answers to your questions:
>
> >Some parts of the experimental paradigm look a bit arbitrary. For instance, the authors use IMDB as an evaluation set. Why that is not clear. Do the results replicate, say, on validation sets of English Wikipedia?
>
>  We used IMDB (and LAMBADA) because we wanted to test robustness on a validation set from a different domain than the one used for training (Wikipedia). That said, the results replicate on other datasets. Results in anonymized link:
>
> https://imgur.com/a/vcwuNWk
>
> We plot the training perplexities for different deletion probabilities for the monolingual and the bilingual model evaluated on English Wikipedia, as suggested by the reviewer.
>
> As shown, the trend is similar to the paper: when we keep all the weights (p=0.0), the monolingual model performs slightly better. However, as we delete more and more weights, the bilingual model finally outperforms the monolingual one. The cross-over point is around deletion probability p=0.2.
>
> We note that the perplexity numbers here are slightly different than the ones for IMDB, we generally observe that both models are more robust. This is expected and intuitive since the models are evaluated in texts from (validation sets) of their training distribution (Wikipedia).
>
>
> >A second experiment uses the LAMBADA dataset as a secondary task. LAMBADA is not a language modeling dataset. It's just an evaluation dataset containing only a few selected passages, so it is not clear why it is a good choice for these experiments.
>
>
> We respectfully disagree with the reviewer on the comment: “LAMBADA is not a language modeling dataset”. In fact, the very name LAMBADA stands for:  LAnguage Modeling Broadened to Account for Discourse Aspects. We used LAMBADA to show that the results we get for the language modeling experiments are not specific to IMDB. We also added an experiment on wikipedia now to further verify robustness.
>
>
> >It is also not clear what is the connection between the attention weights and the predicted effect on L2 regularization.
>
>
> In our theoretical results, we show that for linear representation learning, the robustness to noise addition to the weights depends on the sum of the first k-singular values of the task matrix C (see Theorem 2.5). We then prove that multitasking reduces the norm of the learned Gamma matrix if the tasks are sufficiently diverse, which happens with high probability for random task vectors.
>
> In our experiments for the linear case, we show that indeed multitasking induces weight regularization, see for example Figure 7 in the Appendix and hence real tasks must be sufficiently diverse. An interesting question is whether l2 regularization happens (due to multitasking) in the non-linear case as well. Figure 5 shows that this indeed might be happening. It shows that the maximum singular value in the linear matrices in the attention layers is dropping faster for the bilingual model compared to the monolingual. Of course, this is a very preliminary ablation study and more experiments are needed before we reach any definite answer on what happens to the non-linear case.
>
>
>
> >There seem to be missing references that come up on a quick google search, such as this one: https://arxiv.org/pdf/2004.11072.pdf
>
> We thank the reviewer for the reference, we will make sure to include it in the next revision of our paper. This paper studies robustness to distribution shifts for semantic segmentation. Our work is on how multitasking yields structural robustness to network weight (not data) corruptions and is not specific to any application. We will discuss the paper briefly in our final version.

---

> > ### Author Response · Authors · 2022-08-02
> > **part 2 of reply**
> >
> >
> > >The paper introduces experiments, whose results are only presented in the appendix. This is not appropriate. Either these experiments should be only briefly mentioned in the main body, or the full results should be discussed in it. Only introducing them seems like a way of working around the page limit.
> >
> > Sorry for that -- we are not trying to work around the page limit,  only provide more details.  In any case, the results in the Appendix are mostly ablation studies and they are not so critical for the paper, but we will mention them in the main body too.
> >
> >
> > >It is not clear to me whether the experiments were adequately controlled. Were the "monotask" and "multitask" models trained for a comparable number of iterations on a comparable number of data points?
> >
> > In our experiments, we tried to keep everything controlled including the model architectures and training of both models. The total training data for both the monolingual and the bilingual model was approximately 2GB and both models were trained for 8 epochs. The rest of the training hyperparameters were the same between the two models, see also Section C: Training Details in the Appendix of the paper. We plan to update this section with further details in the next revision of our work.
> >
> > > If so, this is a tad confusing. I would strongly recommend using a different symbol for the transpose
> >
> > Good point, we will use a different letter for the number of tasks.
> >
> > >the authors could revise the title fragment "A Neural Model for Bilingual Cognitive Reserve", which for the uncareful reader could warrant such extrapolations.
> >
> > This is a fair point, but we still think the connection is very interesting and we would prefer to keep it.  We can compromise with “A Simple Model for Bilingual Cognitive Reserve”.

---

> > > ### Comment · Reviewer_WfPh · 2022-08-09
> > > **follow up**
> > >
> > > Thank you for the clarifications. I would just like to follow up on a couple of points:
> > >
> > > 1. I thank the authors for the extra evaluation on Wikipedia. While I celebrate that had explored out-of-domain robustness, I believe it was also a natural question to wonder about in-domain robustness. On the topic of LAMBADA, I would like to ask the authors to clarify when they say that they train on the LAMBADA dataset, to which section of the dataset they refer, and if the amount of text that they use for the wikipedia finetuning is comparable to the small amount that is provided with LAMBADA. LAMBADA is not meant to *train* a language model, but to *evaluate* its accuracy in predicting single words that require processing the broader context. Because it consists of a few excerpts from the Toronto BookCorpus that were human-annotated, the amount of text is relatively small compared to other sources for language modeling. This is what I had in mind in my previous comment. Also, LAMBADA is typically a classification task where the goal is to predict a *single* word in a sentence, so "training" in LAMBADA can be ambiguous between just training on this task or on the standard LM task (which is what I believe the authors did).
> > >
> > > 2. I still don't fully grasp what the authors mean by the "attention weights". Is it the fixed $W_K$ and $W_Q$ matrices from the Transformer that they are analyzing or the result of applying these matrices to the embedding weights (i.e. the "activations")? Both of these would correspond to a "non-linear" case. It seems to me that the authors are analyzing the former, but I would appreciate some more clarity on this regard.

---

> > > > ### Author Response · Authors · 2022-08-09
> > > > **Answer to the follow up questions**
> > > >
> > > > On the first question:
> > > > We again thank the reviewer for recommending the evaluation on Wikipedia, definitely showing in domain robustness strengthens the paper. Regarding LAMBADA: we used this dataset only in the experiment showing decay in the maximum singular value.  We used the whole LAMBADA dataset which is roughly 1GB. We also used 1GB of Wikipedia text. As the Reviewer correctly notices, we trained on the standard LM task (not just predicting the next word). This is worth clarifying, we will update in the next revision of our paper. We thank the reviewer for bringing it up.
> > > >
> > > > On your second question:
> > > > Good question. We corrupt the network matrices (W_Q, W_K, W_V), not the embeddings. We will write this more explicitly in the next revision, thank you.

---

> > > > > ### Comment · Reviewer_WfPh · 2022-08-10
> > > > > **Title**
> > > > >
> > > > > Thank you for the clarifications. One last point regarding the title. You propose "A Simple Model for Bilingual Cognitive Reserve". The issue perhaps is that this title continues to claim that the model you propose *explains* a neuroscientific phenomenon (Bilingual Cognitive Reserve), something for which no direct evidence is provided. Thus, the issue is not with the kind of model (which could be neural or symbolic, simple or complex), but with the "for" part. Thus, I would suggest thinking of alternatives to this preposition such as "inspired by", "based on", "with connections to", etc. Additionally, perhaps around lines 17-18, I would suggest making a link to the limitations described in Appendix E. That would help in making these limitations a bit more prominent and thus, clarify the scope of the paper.
> > > > >
> > > > > That said, I will increase my score from 6 to 7.

---

### Official Review · Reviewer_NDUb · 2022-07-12

**Rating:** 5
**Confidence:** 2
**Soundness:** 3 good
**Presentation:** 3 good
**Contribution:** 2 fair

**Summary:**

This paper investigates the relationship between multitask learning and robustness to neuron failures. The paper mathematically shows that  multitasking models are more robust to noises, and diverse tasks lead to higher robustness than similar tasks. The paper also presents experiments that demonstrate higher robustness in multitasking models (both linguistic and non-linguistic).


**Questions:**

- What would be practical implications and insights?

**Limitations:**

The authors addresses limitations and ethical considerations.

**Strengths And Weaknesses:**

I think the strengths of this paper includes:
- clearly written
- interesting motivation bringing cognitive science and bilingualism into the space
- interesting problem
- experiments are conducted on several datasets and tasks.
- the results could inspire other researchers
- thorough appendix
The weaknesses are:
- the connection between the human and what the paper does sounds interesting, but I cannot fully understand it.
- I'm having trouble in understanding any practical implications and insights.

---

> ### Author Response · Authors · 2022-08-02
> **Specific reply to Rev: NDUb**
>
> Specific reply to Rev: NDUb
> Thank you for your positive and constructive input. Answers to your questions:
>
> >The connection between the human and what the paper does sounds interesting, but I cannot fully understand it.
>
> In Cognitive Neuroscience, it has been observed that bilingualism increases brain robustness by reducing the rate of cognitive decline due to aging and other factors.
>
> On details from neuroscience and cognitive science, please see our response to reviewer QnuC on human brain differences and numerous updated references. We will expand the discussion on these results in the paper too, based on your feedback and the requests of other reviewers.
>
> Inspired by this observation, we show that a similar phenomenon is happening with Artificial Neural Networks. Specifically, we investigate what happens to the performance of a monolingual and a bilingual language model as the weights of the network become more and more corrupted. We show that bilingual neural networks are more robust to different types of corruption to their weights (random deletions, addition of noise and magnitude pruning). Theoretically, we prove that the increased robustness is expected for linear representation learning when there is sufficient task diversity.
>
> As we discuss in our main text, the corruptions we perform to the artificial neural networks can only offer a very simple model for the phenomenon and more research is needed before we can make extrapolations to humans.
>
> >I'm having trouble in understanding any practical implications and insights.
> >What would be practical implications and insights?
>
> Please see the general reply on applications.

---

### Author Response · Authors · 2022-08-02
**General Response**


We thank the reviewers for carefully reading our paper and for providing constructive feedback. We are pleased to see that all the reviewers acknowledged the merits of this work. This is our general rebuttal reply and we further respond to specific reviewer comments as replies to their reviews in these threads. We want to highlight some additional experiments we ran based on reviewer suggestions.

We strengthened the experimental evaluation of our language modeling experiments by measuring robustness on a holdout set from English Wikipedia. This experiment was recommended by Reviewer WfPh, we thank the Reviewer for the suggestion. The Figure summarizing the result is available in the following anonymous link: https://imgur.com/a/vcwuNWk
The results we obtain for English Wikipedia are consistent with the results we have in the paper (where we used IMDB and LAMBADA for evaluation): as we corrupt the weights, the bilingual model is showing higher structural robustness and achieves lower perplexity.

Following Reviewer's 1N4w advice, we also added one more experiment to shed light on what is the is the connection between l2 regularization and the regularization the authors observe by multitasking.

Let's start with some simple intuition from our theorem and two corner case examples:
Consider the matrix C of real task vectors, each of the T columns being a high dimensional vector of the ideal vector for each task.
Assume for simplicity that each column of C has unit norm. As more tasks are added, C gets more columns. Our first result is that robustness depends on the sum of the top k singular values of C and the representation dimension k, stays constant as T grows. We prove that if this sum of the top k singular values grows sublinearly in T, more tasks yield more robustness. Now, if all the task vectors (the unit columns of C) are perfectly aligned, the top singular value of C scales linearly in T (the strongest direction gets more and more energy as tasks are added) and hence we get NO robustness or regularization from multitasking. On the other extreme, if each new task vector is orthogonal, adding new vectors in C does not increase the top-k singular value sum and robustness improves like 1 over T. This hopefully makes it clear why task diversity is needed for robustness, and random tasks are indeed near orthogonal. Our results on real tasks show that real task vectors are also sufficiently diverse. If the reviewers like this example we can add it to the paper.

From this intuition, we hope it is clear that multitasking creates regularization ONLY when the task vectors are near orthogonal, but certainly not if they are near co-linear. On the contrary, explicitly adding L2 regularization is independent of the structure of the task vectors.

Experimentally, we compare the SVD solution and the solution we get by learning with gradient descent in rank k space + L2 regularization. For both SVD and L2 regularized GD solutions, we observe what happens to the MSE as we increase the noise, the amount of regularization, and the number of tasks. The results are shown in the following anonymous links and will be included in the camera-ready version of our paper: SVD solution: https://imgur.com/a/Pm1ru1a GD + L2 solution: https://imgur.com/a/Sgcl9WY. Please see the response to Reviewer 1N4w for further discussion about this experiment.

General response to applications:

Reviewers asked: ‘’What are the applications if indeed artificial neural networks exhibit higher structural robustness by multitask training?’’ Beyond making a toy model for human cognitive reserve and observing an interesting phenomenon, we think there are a few possible applications:
1. The creation of networks that are robust to weight quantization (for lower precision arithmetic implementations and model compression) can benefit from multitasking training robustness.
 2. Finding lottery tickets i.e. training networks that are more suitable for lottery ticket sparsification by benefiting from multitask robustness
3. Understanding how to use multitask representation learning as an implicit regularizer can benefit in designing networks with unreliable hardware or other forms of noisy neural network computation, e.g. with analog circuits.


We emphasize that all three applications are speculative at this point and there is massive volume of literature in all three directions. The only thing we would like to establish here is that the research question we study is interesting and has potential applications for artificial neural networks, beyond human cognitive science.

---

### Meta-Review · Area_Chair_Zn1x · 2022-08-28

**Recommendation:** Accept
**Confidence:** Certain

**Metareview:**

This paper has received positive reviews. There was some initial concern by one reviewer about
* Missing discussion of neuroscience research
* Why noise is added to model weights

But these issues were clarified in the author response and the reviewer has updated their evaluation.

There was additional discussion on the connection to neuroscience research, with one reviewer suggesting not to push the authors to make stronger links than they see fit. The limitations in Appendix E are appropriate and can even feature more prominently in the main body.

In any case, these weaknesses are relatively minor compared to the paper's strengths:
* Interesting motivation and connections between bilingual models and people, and how deficits impact them.
* insightful theoretical analysis
* Experiments with multiple settings

I therefore recommend acceptance.

And in any case, I encourage the authors to take into account all comments by the reviewers when preparing their revision, especially the following:
* Comments and questions by Reviewer WfPh, especially about the experimental paradigm.
* The comment and discussion by Reviewer 1N4w on the theoretical analysis and L2 regularization as a comparison.





**Award:**

No

---

### Decision · Program_Chairs · 2022-09-14

Accept